# List-Decodable Sparse Mean Estimation via Difference-of-Pairs Filtering

**Ilias Diakonikolas**
UW-Madison
ilias@cs.wisc.edu

**Daniel M. Kane**
UC, San Diego
dakane@cs.ucsd.edu

**Sushrut Karmalkar**
UW-Madison
skarmalkar@wisc.edu

**Ankit Pensia**
UW-Madison
ankitp@cs.wisc.edu

**Thanasis Pittas**
UW-Madison
pittas@wisc.edu

## Abstract

We study the problem of list-decodable *sparse* mean estimation. Specifically, for a parameter $\alpha \in (0, 1/2)$, we are given $m$ points in $\mathbb{R}^n$, $\lfloor \alpha m \rfloor$ of which are i.i.d. samples from a distribution $D$ with unknown $k$-sparse mean $\mu$. No assumptions are made on the remaining points, which form the majority of the dataset. The goal is to return a small list of candidates containing a vector $\hat{\mu}$ such that $\|\hat{\mu} - \mu\|_2$ is small. Prior work had studied the problem of list-decodable mean estimation in the dense setting. In this work, we develop a novel, conceptually simpler technique for list-decodable mean estimation. As the main application of our approach, we provide the first sample and computationally efficient algorithm for list-decodable sparse mean estimation. In particular, for distributions with "certifiably bounded" $t$-th moments in $k$-sparse directions and sufficiently light tails, our algorithm achieves error of $(1/\alpha)^{O(1/t)}$ with sample complexity $m = (k \log(n))^{O(t)}/\alpha$ and running time $\mathrm{poly}(mn^t)$. For the special case of Gaussian inliers, our algorithm achieves the optimal error guarantee $\Theta(\sqrt{\log(1/\alpha)})$ with quasi-polynomial complexity. We complement our upper bounds with nearly-matching statistical query and low-degree polynomial testing lower bounds.

## 1 Introduction

It is well-established that when a dataset is corrupted by outliers, many commonly-used estimators fail to produce reliable estimates [Tuk60, ABH+72]. The field of robust statistics was developed to perform reliable statistical inference in the presence of a constant fraction of outliers, even when the data is high-dimensional [HR09, HRRS86]. Although statistical rates of high-dimensional robust estimation problems are relatively well-understood by now [DL88, Yat85, DG92, HR09, CGR16], all of the estimators developed in these works were computationally inefficient, with runtime exponential in the dimension. The goal of algorithmic robust statistics, beginning with the works of [DKK+16, LRV16], is to design computationally efficient algorithms for high-dimensional robust estimation tasks. We refer the reader to the survey [DK19] for an introduction to this field.

The bulk of the recent progress in algorithmic robust statistics has focused on the setting where the fraction of outliers is a small constant, and the majority of samples are inliers, see, e.g., [DKK+16, LRV16, KKM18]. In contrast, when the fraction of outliers outnumbers the fraction of inliers, it is generally information-theoretically impossible to output a single estimate with non-vacuous error guarantees. In such situations, we allow the algorithm to return a small list of candidates such that one of the candidates is close to the true parameter. This *list-decodable* learning setting was first introduced in [BBV08] and developed in [CSV17]. We define the model below.

36th Conference on Neural Information Processing Systems (NeurIPS 2022).

**Definition 1.1** (List-Decodable Learning). *Given a parameter $0 < \alpha < 1/2$ and a distribution family $\mathcal{D}$ on $\mathbb{R}^n$, the algorithm specifies $m \in \mathbb{Z}_+$ and observes a set of $m$ samples constructed as follows: First, a set $S$ of $\lfloor \alpha m \rfloor$ i.i.d. samples are drawn from an (unknown) distribution $D \in \mathcal{D}$. Then, an adversary is allowed to inspect $S$ and choose a multiset $E$ of $m - \lfloor \alpha m \rfloor$ points. The multiset $T$, defined as $T := S \cup E$, of $m$ points is given as input to the algorithm. We say that $D$ is the distribution of inliers, the elements in $S$ are inliers, the points in $E$ are outliers, and $T$ is an $(1 - \alpha)$-corrupted dataset of $S$. The goal is to output a "small" list of hypotheses $\mathcal{L}$ at least one of which is (with high probability) close to the target parameter of $D$.*

The list-decodable learning setting, interesting in its own right, is closely related to several well-studied problems. A natural example is the problem of parameter recovery from mixture models, e.g., Gaussian mixtures (see, e.g., [Das99, VW04, AK05, DS07, KK10, RV17]). List-decodable mean estimation can serve as a key step in learning mixtures, since one can treat any component of the mixture as the set of inliers (see, e.g., [CSV17, DKS18a, KS17]). In addition, list-decodable learning can be used to model data in important applications where mixture models are not sufficient, such as crowdsourcing (see, e.g., [SVC16, SKL17, MV18]) and community detection in stochastic block models (e.g., [CSV17]).

Prior work on list-decodable mean estimation has focused on the unstructured setting, where the target mean is an arbitrary dense vector (see, e.g., [CSV17, KS17, DKS18a, RY20a, CMY20, DKK20, DKK$^+$21, DKK$^+$22b]). Sparse models have proven to be useful in a wide range of statistical tasks, and thus understanding the statistical and computational aspects of sparse estimation is a fundamental problem (see, e.g., [EK12, HTW15, van16]). Here we study list-decodable *sparse* mean estimation, where the target mean vector is known to be $k$-*sparse*, i.e., it has at most $k$ non-zero coordinates. Given an $(1 - \alpha)$-corrupted set of samples, our goal is to output, in a computationally-efficient manner, a small list of vectors containing a good approximation $\widehat{\mu}$ to the true mean $\mu$ (cf. Definition 1.1). Importantly, the goal is to achieve this with *far fewer samples* than in the dense setting. While the dense setting would require sample size polynomial in $n$ — the ambient dimension of the data — the goal here is to solve the problem in number of samples polynomial in $k$ and only polylogarithmic in $n$.

In this paper, we present a novel and conceptually simple technique for list-decodable mean estimation (that is applicable even in the dense setting). Combining our framework with the concentration results from [DKK$^+$22a], we obtain the first sample and computationally efficient algorithm for list-decodable sparse mean estimation. We note that, while prior results [KS17, RY20a] can possibly be modified to incorporate the sparsity framework of [DKK$^+$22a] after sufficient effort, a notable contribution of our work is a general and conceptually simpler framework for list-decodable estimation, which can easily be adapted to incorporate various structural constraints.

## 1.1 Related Work

Efficient estimators for high-dimensional robust statistics with sparsity constraints have been recently developed for various problems, such as mean estimation and PCA (see, e.g., [BDLS17, DKK$^+$19, Li17, DKK$^+$22a]). The problem of list-decodable mean estimation was first introduced in [CSV17], which achieved an error guarantee of $\tilde{O}(1/\sqrt{\alpha})$ for distributions with bounded second moment; this guarantee turned out to be optimal for this distributional assumption (see [DKS18a]). Subsequent work improved the algorithmic guarantees for this problem (see, e.g., [CMY20, DKK20, DKK$^+$21]).

To achieve better error guarantees, it is necessary to make further assumptions on the distribution, e.g., Gaussianity or bounded higher moments. In terms of the minimax optimal rate[1], [DKS18a] showed that the optimal error for Gaussians is $\Theta(\sqrt{\log(1/\alpha)})$. They also showed that any SQ algorithm that achieves the optimal error for Gaussians must take either super-polynomial time or samples, and presented an algorithm with matching guarantees. When the distribution $D$ has bounded $t$-th moment for some even $t > 2$ ( i.e., $\mathbf{E}[\langle v, X - \mathbf{E}[X] \rangle^t]$ is bounded for all unit vectors $v$), [DKS18a] proved that the minimax optimal rate is $\Theta(\alpha^{-1/t})$. For distributions with certifiably bounded $t$-th moments, [KS17, RY20a] provided algorithms obtaining an error rate of $O((1/\alpha)^{O(1/t)})$ with sample complexities $m = \text{poly}(n^t/\alpha)$, and runtimes $\text{poly}(m^t n^t)$ and $\text{poly}(mn)^{\text{poly}(t,1/\alpha)}$, respectively. Recently, in the context of moment estimation and clustering

---

[1]Informally speaking, we say that the minimax optimal rate is $\gamma$ if (i) no algorithm (regardless of sample size and runtime) has error $o(\gamma)$ with a list of size independent of dimension $n$, and (ii) there is an algorithm with error $O(\gamma)$ with a list of size independent of $n$; in our case, $\text{poly}(n/\alpha)$ samples and $O(1/\alpha)$ list size suffice.

problems, [ST21] showed how to improve the dependence on $m$ in the runtime of algorithms that depend on certifiably bounded moments from $\mathrm{poly}(m^t n^t)$ to $\mathrm{poly}(mn^t)$. While it is possible that their result might be applied to [KS17, RY20a], our algorithmic technique naturally lends itself to achieve runtime $\mathrm{poly}(mn^t)$ for the dense as well as sparse settings. We provide detailed comparisons with (see, e.g., [DKS18a, KS17, RY20a]) in Section 1.3. Finally, we mention that the list-decodable setting has also been studied in the context of linear regression (see, e.g., [KKK19, RY20a, DKP⁺21]) and subspace recovery (see, e.g., [BK21, RY20b]).

In an independent and concurrent work [ZS22], Zeng and Shen also study list-decodable sparse mean estimation when the underlying distribution is spherical Gaussian.

## 1.2 Our Results

We demonstrate an algorithm to perform list-decodable sparse mean estimation with $(k \log n)^{O(t)}$ samples, when the mean $\mu \in \mathbb{R}^n$ is known to be $k$-sparse. For this to be possible, we will require some assumptions on the underlying distribution of inliers $D$. Prior work in the dense setting ([KS17, RY20a]) assumed that the inlier distribution $D$ in Definition 1.1 satisfies $d$-*certifiably* bounded $t$-th moments in every direction (i.e., for some moment bound $M > 0$, $M\|v\|_2^t - \mathbf{E}_{X \sim D}[\langle v, X - \mu \rangle^t]$ can be expressed as a sum of square polynomials of degree at most $d = O(t)$ in the entries of $v$), and $D$ has light tails. We highlight that our algorithmic technique can also be used in the dense case, under the same assumption, and provides qualitatively similar error guarantees with much simpler arguments and improved runtime. Below we apply our technique to the sparse setting.

Our results hold when (i) the $t$-th moment of $D$ is $d$-certifiably bounded for every $v$ that is $k$-sparse, with $d = O(t)$, and (ii) $D$ has light tails. For ease of exposition, we state the result assuming $D$ has subexponential tails (i.e., for some universal constant $c$, for all unit vectors $v$ and $p \in \mathbb{N}$, $\mathbf{E}_{X \sim D}[|\langle v, X - \mu \rangle|^p]^{1/p} \le cp$). [2]

**Theorem 1.2** (List-Decodable Sparse Mean Estimation). *Let $t$ be an integer power of two. Let $D$ be a distribution over $\mathbb{R}^n$ with $k$-sparse mean $\mu$. Suppose that $D$ has $t$-th moments $d$-certifiably bounded in $k$-sparse directions by $M$ for some $d = O(t)$ (cf. Definition 2.3) and subexponential tails in the standard basis directions. There is an algorithm which, given $\alpha$, $M$, $t$, $k$, and a $(1-\alpha)$-corrupted set of $m = (tk \log n)^{O(t)} \max(1, M^{-2})/\alpha$ samples from $D$, runs in time $\mathrm{poly}(mn^t)$ and returns a vector $\hat{\mu} \in \mathbb{R}^n$ such that with probability $\Omega(\alpha)$ it is the case that $\|\hat{\mu} - \mu\|_2 = O_t(M^{1/t}/\alpha^{O(1)/t})$.*

Note that with high probability over the inliers and for any choice of outliers, with probability $\Omega(\alpha)$ over the internal randomness of the algorithm, the algorithm of Theorem 1.2 outputs an estimate $\hat{\mu}$ close to $\mu$. By running our algorithm $O(1/\alpha)$ times, we can generate a list of size $O(1/\alpha)$ such that with probability $0.9$ the list contains the desired estimate $\hat{\mu}$.

Notably, for the important special case of Gaussian $\mathcal{N}(\mu, I)$ inliers, our algorithm achieves the information-theoretically optimal error rate. This is because $\mathcal{N}(\mu, I)$ has its $t$-th moment certifiably bounded by $t^{t/2}$ in all directions. Specifically, for a large enough constant $C > 0$, we obtain the following result: Given $\alpha$, $t$, and a $(1 - \alpha)$-corrupted set of $m \ge (tk \log n)^{Ct}$ samples from $\mathcal{N}(\mu, I)$ for a $k$-sparse vector $\mu$, our algorithm runs in time $\mathrm{poly}(mn^t)$ and with probability $\Omega(\alpha)$ outputs a vector $\hat{\mu}$ such that $\|\hat{\mu} - \mu\|_2 \le O(\sqrt{t}/\alpha^{C/t})$. Thus, by taking $t = C \log(1/\alpha)$, we obtain the optimal error of $\Theta(\sqrt{\log(1/\alpha)})$ in quasi-polynomial sample and time complexity.

We also note that a broad and natural class of distributions satisfying Definition 2.3 is the class of $\sigma$-Poincare distributions (see, e.g., [KS17]). A distribution is said to be $\sigma$-Poincare if for all differentiable functions $f : \mathbb{R}^n \to \mathbb{R}$, we have that $\mathbf{Var}_{X \sim D}[f(X)] \le \sigma^2 \mathbf{E}_{X \sim D}[\|\nabla f(X)\|_2^2]$.

We complement our algorithm of Theorem 1.2 with a qualitatively matching lower bound in the Statistical Query (SQ) model [Kea98]. Instead of directly accessing samples, SQ algorithms are only allowed to perform adaptive queries of expectations of bounded functions of the underlying distribution, up to some desired tolerance (c.f. Definition C.1). The class of SQ algorithms is fairly broad: a wide range of known algorithmic techniques in machine learning are known to be implementable in the SQ model (see, e.g., [FGR⁺13]).

An SQ lower bound is an unconditional statement that for any SQ algorithm, either the number of queries $q$ must be large or the tolerance, $\tau$, of some query must be small. Since simulating a query of

---

[2]It suffices for $D$ to have $\mathrm{poly}(t \log(n))$ bounded moments in axis-aligned directions; see Section 2.

tolerance $\tau$ by averaging i.i.d. samples may need up to $\Omega(1/\tau^2)$ many of them, SQ lower bounds are naturally interpreted as a tradeoff between runtime $\Omega(q)$ and sample complexity $\Omega(1/\tau^2)$. An adaptation of the result in [DKS18a] yields Theorem 1.3, which indicates that the $k^{O(t)}$ factor in the sample complexity of Theorem 1.2 might be necessary for efficient algorithms, even for Gaussian inliers.

**Theorem 1.3** (SQ Lower Bound, Informal). *Consider the problem of list-decoding the mean of $\mathcal{N}(\mu, I)$, for a $k$-sparse vector $\mu \in \mathbb{R}^n$, up to error better than $O((t\alpha)^{-1/t})$. Any SQ algorithm that solves the problem does one of the following: (i) It returns a list of size $n^{k^{\Omega(1)}}$, (ii) it uses at least one query of tolerance $k^{-\Omega(t)} \exp(O(t\alpha)^{-2/t})$, or (iii) it makes at least $n^{k^{\Omega(1)}}$ queries.*

Theorem 1.3 states that any SQ algorithm that runs in time less than $n^{k^c}$ for a constant $c > 0$ and achieves the minimax-optimal error of $\sqrt{\log(1/\alpha)}$ must use at least $k^{\Omega(\log(1/\alpha)/\log\log(1/\alpha))}$ samples. This follows by setting $t = \Omega(\log(1/\alpha)/\log\log(1/\alpha))$. A similar lower bound holds for the computational model of low-degree polynomial tests, as a consequence of the recently established relationship between the two models [BBH+21]. See Appendix C for more details.

## 1.3 Overview of Techniques

We begin with a brief overview of the existing techniques for *dense* list-decodable mean estimation. We then highlight some of the obstacles in adapting these techniques to the sparse setting. Then, we provide an overview of our algorithmic approach and explain how it overcomes these obstacles.

**Prior Work on List-Decodable Mean Estimation** In the dense setting, prior algorithmic techniques for list-decoding the mean with error better than $\Omega(\alpha^{-1/2})$ are quite complicated. [DKS18a] uses a multifilter-based technique for list-decoding spherical Gaussians, which relies critically on knowing the higher degree moments of the inliers (and thus does not generalize to less specific distribution families). Moreover, this method runs into technical difficulties related to being unable to determine the variance of higher degree polynomials on the inliers without knowing the mean ahead of time. The other approach in the literature (see, for example, [KS17, RY20a]) uses the Sum-of-Squares method (SoS) to find these clusters of points. The algorithm in [RY20a] involves solving a nested SoS program and then applying a complicated rounding procedure to get the final list. It should be noted that the runtime of [RY20a] is exponential in $\text{poly}(1/\alpha)$, which can be quite large. Finally, [KS17] gave an SoS based list-decodable mean estimation algorithm for the *dense* case with error, sample complexity, and list size similar to the ones that are obtained in our work; but significantly worse runtime. The approach of [KS17] has some important differences. First, the clustering relaxation is conceptually harder, involving a more complex optimization problem for each filtering step and second, after the filtering ends, the error guarantee scales with the norm of the unknown mean; thus a complicated re-clustering step that combines ideas from [SCV18] is needed to reduce the error.

Here we present a significantly cleaner method to perform the outlier removal step; we avoid problems like not knowing the mean ahead of time by simply taking pairs of differences of our samples to make their mean zero. While it is likely that either of the prior techniques could be adapted to the sparse setting with sufficient effort, this would result in significantly more complicated algorithms. We briefly point out some difficulties below.

First, to ensure that the algorithms identify subsets of the samples that satisfy certifiably bounded moments in all $k$-sparse directions, requires additional variables and constraints to the algorithms. Additionally, one would need to replace the bounded moments in all directions condition by the corresponding condition for the sparse case, and ensure that all the proof steps can be modified to rely only on the latter – this would result in minor modifications of the original algorithms, such as thresholding of candidate solution vectors.

Second, at the end of this process, while the algorithm might qualitatively match the error guarantee that we achieve, the runtime would continue to be $(mn)^{O(t)}$ for [KS17] or $(1/\alpha)^{\text{polylog}(1/\alpha)} n^{O(\max\{1/\alpha^4, t\})}$ for [RY20a] — both of these are qualitatively worse than the runtime we achieve when $\alpha$ is sufficiently small. To obtain improved runtime using these prior techniques, one would require additional ideas, e.g., from [ST21], to be adapted to this setting, overall resulting in a far more complicated algorithm.

On the other hand, it seems difficult to adapt the multi-filtering technique from [DKS18a] to the setting we consider, without any introduction of an SoS component. We remind the reader that the [DKS18a]

algorithm depends critically on knowing the higher moments of the inliers *exactly*, and does not generalize to less specific distribution families. Even in the Gaussian setting, generalizing [DKS18a] might be difficult, since it would require the design of an efficiently verifiable notion of matching higher moments in $k$-sparse directions.

We believe that our novel list-decoding technique is significantly simpler. As a result of simplifying the optimization programs involved, our technique naturally improves the runtime from $\mathrm{poly}(m^t n^t)$ in prior work to $\mathrm{poly}(mn^t)$.

**Novel List-Decodable Mean Estimation Algorithm** All known list-decoding algorithms are based upon the following fundamental observation. Suppose that $S$ is a dataset that contains a subset $S_\mathrm{g}$ of samples with bounded moments. If we can find a subset $S'$ of $S$ with bounded moments and large overlap with $S_\mathrm{g}$, then the means of $S'$ and $S_\mathrm{g}$ cannot be too far apart (see Lemma 3.3). The goal of a list-decoding algorithm is to find such a subset $S'$ (or, more precisely, a small number of hypotheses for such a subset). In order to generalize this to the sparse mean setting, it suffices for the subsets $S_\mathrm{g}$ and $S'$ to have bounded moments only in all $k$-sparse directions. This will imply that $|\langle v, \mu_\mathrm{g} - \mu_{S'} \rangle|$ is relatively small for all $k$-sparse unit vectors $v$, which will in turn imply that truncating $\mu_{S'}$ to its $k$ largest entries will provide a suitable approximation to $\mu_\mathrm{g}$ (see Fact A.2).

The basic idea behind our novel list-decoding technique, which we call *the difference of pairs filter*, is the following: let $T$ be the set of differences of pairs of elements of $S$. The set $T$ will contain a relatively large subset, $T_\mathrm{g}$, (consisting of the pairwise differences of elements of $S_\mathrm{g}$) whose moments in $k$-sparse directions are bounded. Our goal will be to find a subset $T' \subset T$ that has large overlap with $T_\mathrm{g}$ and also has bounded ($k$-sparse) moments.

Naïvely, we can do this as follows. We start with $T' = T$. Either this set has bounded $k$-sparse moments (in which case we are done) or there is some sparse direction $v$ in which the average value of $|\langle v, x \rangle|^t$ over $T'$ is substantially larger than the average value over $T_\mathrm{g}$. By throwing away points $x$ from $T'$ with probability proportional to $|\langle v, x \rangle|^t$, we eliminate mostly bad points. We repeat this until $T'$ has bounded sparse moments. Unfortunately, while this approach can be shown to be correct, it does not suffice for our purposes because it is computationally infeasible in general to determine whether or not $T'$ has bounded sparse moments. However, if we assume additionally that $S_\mathrm{g}$ has bounded $k$-sparse moments *provable by a low-degree sum-of-squares proof* (i.e., the moment bound inequality can be re-expressed as a sum of square polynomials being greater than $0$), then so will $T_\mathrm{g}$. There is an efficient algorithm to determine whether or not $T'$ has certifiably bounded moments in $k$-sparse directions as well. If $T'$ does not have certifiably bounded moments, standard techniques for Sum-of-Squares programs imply that we can manufacture a non-negative polynomial $p$ so that the average value of $p$ on $T'$ is substantially larger than the value of $p$ on any set with SoS-certifiable bounded moments in $k$-sparse directions. Thus, filtering out points $x$ in $T'$ with probability proportional to $p(x)$ will likely remove mostly bad points. Using this idea, we can find such a set $T'$ efficiently (see Theorem 3.1).

We are thus left with a set of differences of samples rather than a set of samples. At this point, we will need a rounding method that given a set $T'$ of differences with bounded $k$-sparse moments guaranteed to have large overlap with $T_\mathrm{g}$, finds a small list of sets $S'$ with bounded $k$-sparse moments so that at least one of them has a large overlap with $S_\mathrm{g}$. Note that $T'$ might be the union of $T_i - T_i$, where each $T_i$ is an individual cluster drawn from a moment-bounded distribution. This demonstrates the necessity of a rounding step to identify the means of individual clusters. To achieve this, it is helpful to think of $T'$ as consisting of a set of pairs of elements of $S$, or equivalently as a graph over $S$. Given $T'$, our first task is to find some reasonably large subset of $S$ with bounded moments (ideally which has a large overlap with $S_\mathrm{g}$). It is not hard to see that it suffices to find any large clique in $T'$ (see Lemma 3.3). Unfortunately, $T'$ may well not have any large cliques, and even if it does, finding them may be computationally difficult. However, we are saved here by the observation that if $|\langle v, x - y \rangle|^t$ and $|\langle v, y - z \rangle|^t$ are both small, then so is $|\langle v, x - z \rangle|^t$. This means that if we replace $T'$ by the graph $H$ of all pairs of vertices $(v, w)$, where $v$ and $w$ have many common neighbors in $T'$, this new graph will **also** have relatively small $k$-sparse moments (see Lemma 3.5). As most elements of $S_\mathrm{g}$ are adjacent in $T'$ to most other elements of $S_\mathrm{g}$, it is not hard to see that this new graph will have relatively large cliques; unfortunately, finding them may still be computationally difficult.

To find these large cliques efficiently, we need one final observation. If $v$ is a random vertex of a graph $G$, then there will (on average) be very few pairs of neighbors, $u$ and $w$, of $v$ so that $u$ and $w$ do not have a large number of common neighbors with each other in $G$ (see Lemma 3.6). This means

that if we pick a random sample $x$ in $S$, then most pairs of neighbors of $x$ in $T'$ are neighbors in $H$. Using a densification procedure (see Lemma 3.7), it is not hard to find a large subset $S'$ of these neighbors so that any two elements of $S'$ have many neighbors in common in $G$. By Lemma 3.5, this implies that $S'$ will in fact have bounded $k$-sparse moments. Furthermore, if we happened to pick $x$ from $S_{\mathrm{g}}$, it is not hard to see that it is likely that $S'$ has large overlap with $S_{\mathrm{g}}$, and thus its mean provides us with a good estimate.

While the above describes our algorithmic approach, we also need to consider the sample complexity of our method. We know that the distribution $D$ has the property of bounded moments in $k$-sparse directions, and our algorithm requires that the property is also satisfied by the uniform distribution over the samples $S_{\mathrm{g}}$. In order for $S_{\mathrm{g}}$ to have it as well, it suffices that the $t^{th}$ moment tensor of $S_{\mathrm{g}} - \mu$ be close to the corresponding moment tensor of $D - \mu$. It turns out that if these moment tensors are $\delta k^{-t}$-close coordinate-wise, which takes $O(t \log(n) k^{O(t)} / \delta^2)$ samples, this suffices to get the kind of certifiable concentration we require. This is captured by Lemma 2.4, a restatement from [DKK+22a].

## 2 Preliminaries

**Basic Notation.** We use $\mathbb{Z}_+$ to denote positive integers. For $n \in \mathbb{Z}_+$ we denote $[n] := \{1, \ldots, n\}$. We denote by $\mathbb{R}[x_1, \ldots, x_n]_{\leq d}$ the set of real-valued polynomials of degree at most $d$ in variables $x_1, \ldots, x_n$. We use $\mathrm{poly}(\cdot)$ to indicate a quantity that is polynomial in its arguments. For an ordered set of variables $Q = \{x_1, \ldots, x_n\}$, we will denote $p(Q)$ to mean $p(x_1, \ldots, x_n)$. We use $I_n$ to denote the $n \times n$ identity matrix. For a vector $v$, we let $\|v\|_2$ denote its $\ell_2$-norm. We call a vector $k$-sparse if it has at most $k$ non-zero coordinates. We use $\langle v, u \rangle$ for the inner product of the vectors $u, v$. We use capital letters for random variables and $\mathbf{E}[\cdot]$ for expectation. When $S$ is a set, we will use the notation $X \sim S$ to mean that $X$ is distributed uniformly over $S$. For a graph $G$ we denote by $N_G(x)$ the neighborhood of $x$ in $G$. Throughout the paper, we will use the letter $n$ for the dimension, $m$ for the number of samples, $d$ for the degrees of the SoS proofs, and $t$ for the number of bounded moments.

**SoS Preliminaries** The following preliminaries are specific to the SoS part of this paper. We refer to [BS16, FKP19] for a more complete treatment of the SoS framework. Here, we review the basics.

**Definition 2.1** (SoS Proof). *Let $x_1, \ldots, x_n$ be indeterminates and let $\mathcal{A}$ be a set of polynomial equalities $\{p_1(x) = 0, \ldots, p_m(x) = 0\}$. An SoS proof of the inequality $r(x) \geq 0$ from axioms $\mathcal{A}$ is a set of polynomials $\{r_i(x)\}_{i \in [m]} \cup \{r_0(x)\}$ such that $r_0$ is a sum of square polynomial and $r_i$'s are arbitrary, and $r(x) = r_0(x) + \sum_{i \in [m]} r_i(x) p_i(x)$. If the set of polynomials $\{r_i(x) \cdot p_i(x) \mid i \in [d]\} \cup \{r_0(x)\}$ have degree at most $d$, we say that this proof is of degree $d$ and denote it by $\mathcal{A} \vdash_d r(x) \geq 0$. When we need to emphasize what indeterminates are involved in a particular SoS proof, we denote it by $\mathcal{A} \vdash_d^x r(x) \geq 0$. When $\mathcal{A}$ is empty, we omit it, e.g., $\vdash_d r(x) \geq 0$ or $\vdash_d^x r(x) \geq 0$.*

We will also use the objects called *pseudoexpectations*.

**Definition 2.2** (Pseudoexpectation). *Let $x_1, \ldots, x_n$ be indeterminates. A degree-$d$ pseudoexpectation $\tilde{\mathbf{E}}$ is a linear map $\tilde{\mathbf{E}} : \mathbb{R}[x_1, \ldots, x_n]_{\leq d} \to \mathbb{R}$ from degree-$d$ polynomials to $\mathbb{R}$ such that $\tilde{\mathbf{E}}\left[p(x)^2\right] \geq 0$ for any $p$ of degree at most $d/2$ and $\tilde{\mathbf{E}}[1] = 1$. If $\mathcal{A} = \{p_1(x) = 0, \ldots, p_m(x) = 0\}$ is a set of polynomial inequalities, we say that a pseudoexpectation $\tilde{\mathbf{E}}$ satisfies $\mathcal{A}$ if for every $i \in [m]$, $\tilde{\mathbf{E}}[s(x)p_i(x)] = 0$ for all polynomials $s(x)$ such that $s(x)p_i(x)$ has degree at most $d$.*

It is well known (see, e.g., [BS16]) that pseudoexpectations are dual objects to SoS proofs in the following sense: given a set $\mathcal{P}$ of $r$ polynomial equalities in $n$ variables and a polynomial $q(x_1, \ldots, x_n)$, either there exists an SoS proof $\mathcal{P} \vdash_\ell^x q(x) \geq 0$ or there exists a pseudoexpectation $\tilde{\mathbf{E}}$ of degree $\ell$ satisfying $\mathcal{P}$ but having $\tilde{\mathbf{E}}[q(x)] < 0$. More importantly, there is an algorithm that runs in time $(rn)^{O(\ell)}$ and finds that pseudoexpectation when we are in the second case. Due to lack of space, we defer formal statements and a brief overview of SoS to Appendix A.1.

**Certifiably Bounded Moments in $k$-sparse Directions** Our algorithm succeeds whenever the uncorrupted samples have *certifiably bounded moments* in $k$-sparse directions, defined as in [DKK+22a]:

**Definition 2.3** (($M, t, d$) Certifiably Bounded Moments in $k$-sparse Directions). *Let $Q := \{v_1, \ldots, v_n, z_1, \ldots, z_n\}$ and $\mathcal{A}_{k\text{-sparse}} := \{z_i^2 = z_i\}_{i \in [n]} \cup \{v_i z_i = v_i\}_{i \in [n]} \cup \{\sum_{i=1}^n z_i = k\} \cup$*

$\left\{\sum_{i=1}^n v_i^2 = 1\right\}$. *For an $M > 0$ and even $t \in \mathbb{N}$, a distribution $D$ with mean $\mu$ satisfies $(M, t, d)$ certifiably bounded moments in $k$-sparse directions if $\mathcal{A}_{k\text{-sparse}} \left|\frac{Q}{d} \; \mathbf{E}_{X \sim D} \left[\langle v, X - \mu \rangle^t\right]^2 \leq M^2\right.$.*

The definition of $\mathcal{A}_{k\text{-sparse}}$ is based on the fact that a vector $v = (v_1, \ldots, v_n)$ is $k$-sparse if and only if there exists $z = (z_1, \ldots, z_n)$ such that $v, z$ satisfy $\mathcal{A}_{k\text{-sparse}}$.

We will use the following lemma proved in [DKK+22a] to bound the number of samples it takes to certify bounded moments in $k$-sparse directions. Although this is stated for subexponential distributions, it needs the distribution to have only bounded $t^2 \log(d)$ moments (see Lemma A.10).

**Lemma 2.4** ([DKK+22a]). *Let $D$ be a distribution over $\mathbb{R}^n$ with mean $\mu$. Suppose that $D$ has $c$-sub-exponential tails around $\mu$ for a constant $c$ and that $\mathcal{A}_{k\text{-sparse}} \left|\frac{Q}{O(t)} \; \mathbf{E}_{X \sim D} \left[\langle v, X - \mu \rangle^t\right]^2 \leq M^2\right.$. Let $S = \{X_1, \ldots, X_m\}$ be a set of $m$ i.i.d. samples from $D$, $D'$ be the uniform distribution over $S$, and $\overline{\mu} := \mathbf{E}_{X \sim D'}[X]$. If $m > C(tk(\log n))^{5t} \max(1, M^{-2})$ for a sufficiently large constant $C$, then with probability at least $0.9$ we have the following: (i) $\mathcal{A}_{k\text{-sparse}} \left|\frac{Q}{O(t)} \; \mathbf{E}_{X \sim D'} \left[\langle v, X - \overline{\mu} \rangle^t\right]^2 \leq 8M^2\right.$ and (ii) $\langle v, \overline{\mu} - \mu \rangle \leq M^{1/t}/\alpha^{6/t}$ for every $k$-sparse unit vector $v$.*

# 3 Main Result: Proof of Theorem 1.2

Recall our setting: we are given $\alpha \in (0, 1/2)$ and a multiset $S := \{x_1, \ldots, x_m\}$ such that an unknown subset of $\lfloor \alpha m \rfloor$ many of these points satisfy $(M, t, d)$ certifiably bounded central moments in $k$-sparse directions, and the remaining are arbitrary. The goal is to recover a candidate that is close to $\mu := \mathbf{E}_{X \sim D}[X]$ with probability $\Omega(\alpha)$. In what follows, $t$ will always be $2^\ell$ for some $\ell \in \mathbb{Z}_+$.

## 3.1 The SoS-based Filter

Let $T$ be the set of *pairwise differences* of all samples in $S$ (similarly denote by $T_g$ the subset of $T$ corresponding to inliers $S_g$). We would like to either detect that the moments of $T$ are already bounded in all $k$-sparse directions or find a direction that violates this and filter out mostly outliers. Unfortunately, this kind of check is computationally infeasible, but it can be done efficiently if we check for moment bounds that are certified by SoS proofs. This is done in Algorithm 1, which takes as input the set $T$ along with the parameters $t, d, M$ and performs filtering until the resulting set $T'$ has $(M, t, d)$-bounded moments.

---

**Algorithm 1** SoS-based filter for list-decodable mean estimation

1: **function** LDMEAN-SOS-FILTER($T := \{x_1, \ldots, x_m\}, t, d, M$)
2:     Let $Q = \{v_1, \ldots, v_n, z_1, \ldots, z_n\}$ and $T' = T$.
3:     **while** there is no SoS proof of $\mathcal{A}_{k\text{-sparse}} \left|\frac{Q}{d} \sum_{x \in T'} \langle v, x \rangle^t \leq 6M|T|\right.$ **do**
4:         Find a degree-$d$ $\tilde{\mathbf{E}}$ on $Q$ satisfying $\mathcal{A}_{k\text{-sparse}}$ and $\tilde{\mathbf{E}}[\sum_{x \in T} \langle v, x \rangle^t] > 6M|T|$.
5:         Throw out $x \in T'$ with probability $\tilde{\mathbf{E}}\left[\langle v, x \rangle^t\right] / \max_{x \in T'} \tilde{\mathbf{E}}\left[\langle v, x \rangle^t\right]$.
6:     **end while**
7:     **return** $T'$
8: **end function**

---

**Theorem 3.1** (Filter Identifies a Subset Satisfying Bounded Moments). *Let $T$ be a multiset of points in $\mathbb{R}^n$ for which there exists a subset $T_g \subset T$ with $|T_g| = \alpha^2 |T|$ for some $\alpha > 0$. Furthermore assume that $T_g$ has zero mean and $(M, t, d)$-certifiably bounded moments in $k$-sparse directions for some $M > 0, d \in \mathbb{Z}_+$, and even $t$. Then Algorithm 1, given $T, M, t, d$, returns a subset $T' \subseteq T$ in time $\text{poly}(mn^d)$ so that, with probability at least $2/3$, the following holds: (i) For any $k$-sparse unit vector $v$, we have that $\sum_{x \in T'} \langle v, x \rangle^t \leq 6M|T|$, and (ii) $|T' \cap T_g| \geq |T_g|/2$.*

*Proof.* Let $Q = \{z_1, \ldots, z_n, v_1, \ldots, v_n\}$. In Algorithm 1, Lines 3 and 4 use the separation oracle of Theorem A.4 with $\mathcal{P} = \mathcal{A}_{k\text{-sparse}}$. Since $T_g$ has zero mean and $(M, t, d)$ bounded central moments in $k$-sparse directions, we have that

$$\mathcal{A}_{k\text{-sparse}} \left|\frac{Q}{d} \; M - \mathop{\mathbf{E}}_{X \sim T_g} \left[\langle v, X \rangle^t\right] \geq 0.\right. \tag{1}$$

Thus, if $\mathcal{A}_{k\text{-sparse}} \left| \frac{Q}{d} \sum_{x \in T'} \langle v, x \rangle^t \leq 6M|T| \right.$ the algorithm identifies that and stops (in which case we have the desired conclusion that for any $k$-sparse vector $v$, $\sum_{x \in T'} \langle v, x \rangle^t \leq 6M|T|$); otherwise it finds a degree-$d$ pseudo-expectation $\tilde{\mathbf{E}}$ on $Q$ satisfying $\mathcal{A}_{k\text{-sparse}}$ and $\tilde{\mathbf{E}} \left[ -6M|T| + \sum_{x \in T'} \langle v, x \rangle^t \right] > 0$, in which case we can create a filter: Using Equation (1) and $|T_{\mathrm{g}}| = \alpha^2 |T|$, we have that $M \geq \mathbf{E}_{X \sim T_{\mathrm{g}}} \left[ \tilde{\mathbf{E}}[\langle v, X \rangle^t] \right] \geq \frac{\alpha^{-2}}{|T|} \sum_{x \in T_{\mathrm{g}} \cap T'} \tilde{\mathbf{E}} \left[ \langle v, x \rangle^t \right]$. Since $\tilde{\mathbf{E}} \left[ -6M|T| + \sum_{x \in T'} \langle v, x \rangle^t \right] > 0$ and $\tilde{\mathbf{E}}$ is linear, we see that $\frac{\sum_{x \in T_{\mathrm{g}} \cap T'} \tilde{\mathbf{E}}[\langle v, x \rangle^t]}{\sum_{x \in T'} \tilde{\mathbf{E}}[\langle v, x \rangle^t]} \leq \alpha^2/6$. This means if we throw out sample $x$ with probability $\tilde{\mathbf{E}} \left[ \langle v, x \rangle^t \right] / \max_{x \in T'} \tilde{\mathbf{E}} \left[ \langle v, x \rangle^t \right]$ (which is indeed in $[0, 1]$ since $\langle v, x \rangle^t$ is SoS, so its pseudoexpectation is a non-negative value), on average, only an $\alpha^2/6$ fraction of the points that are removed will be from $T_{\mathrm{g}}$. Since the $x$ with the largest value of $\tilde{\mathbf{E}} \left[ \langle v, x \rangle^t \right]$ will be removed, the algorithm will terminate in polynomial time. We now analyze the size of $|T' \cap T_{\mathrm{g}}|$. By the above analysis, at each step the expected number of samples thrown out from $T_{\mathrm{g}}$ is at most $\alpha^2/6$ times the expected total number of samples removed. Thus, the potential function $\Delta = (|T_{\mathrm{g}} \cap T'| - (\alpha^2/6)|T'|)/|T|$ is a submartingale. Note that always $0 \leq \Delta/\alpha^2 \leq 1$, and initially we had $\Delta/\alpha^2 \geq 5/6$. By Doob's martimgale inequality (Proposition A.9 applied with $t = 1/2$ to the submartingale $(\Delta/\alpha^2)$), the probability that $\Delta/\alpha^2$ remains at least $\alpha^2/2$ throughout the execution of the algorithm is at least $2/3$. Thus, we will have $|T_{\mathrm{g}} \cap T'| \geq (\alpha^2/2)|T| = |T_{\mathrm{g}}|/2$ throughout the execution. $\qquad \square$

## 3.2 Identifying a Subset of Samples with Bounded Moments

Having identified a subset $T' \subset T$ satisfying the conclusions of Theorem 3.1, we want to extract from $T'$ a vector that is close to the original mean. Since the average of the set of differences is likely to be close to zero regardless of the true mean, we will need to use the information about the pairs that we get from $T'$ to find subsets of the original samples that satisfy the appropriate concentration bounds. We will need the following definition.

**Definition 3.2.** *Let $S \subset \mathbb{R}^n$. A graph $(V, E)$ on $S$ with $V = S$ is said to have $(M, t)$- bounded moments in $k$-sparse directions if for all $k$-sparse unit vectors $v$, $(1/|S|^2) \sum_{(x,y) \in E} \langle v, x - y \rangle^t \leq M$.*

By the guarantee of our filter, if $T'$ is the set returned, the graph $G$ with edges $(x, y)$ for which $x - y$ or $y - x$ belongs to $T'$ will have bounded moments in the above sense. If $G$ contains a clique $C$ which intersects with an $\alpha$-fraction of the target samples $C_{\mathrm{g}}$, the means of $C$ and $C_{\mathrm{g}}$ are close.

**Lemma 3.3.** *Let $S \subset \mathbb{R}^n$ and $G$ be a graph on $S$ satisfying Definition 3.2. Let $C \subset S$ be a clique in $G$. Let $C_g \subset C$ be a subset with $|C_g| \geq \alpha|S|$. If $\mu_C$ and $\mu_g$ denote the means of $C$ and $C_g$ respectively, then $\langle v, \mu_C - \mu_g \rangle^t \leq 2M/\alpha^2$ for all $k$-sparse unit vectors $v$.*

*Proof.* By using the fact that $t$ is even and the fact that $G$ satisfies Definition 3.2, we see that $M|S|^2 \geq \sum_{(x,y) \in E} \langle v, x - y \rangle^t \geq \frac{1}{2} \sum_{x,y \in C} \langle v, x - y \rangle^t \geq \frac{1}{2} \sum_{x \in C_g, y \in C} \langle v, x - y \rangle^t$. Using Jensen's inequality we obtain, $M|S|^2 \geq \sum_{x \in C_g, y \in C} \frac{\langle v, x - y \rangle^t}{2} \geq \frac{|C_g||C|\langle v, \mu_C - \mu_g \rangle^t}{2} \geq \frac{|S|^2 \alpha^2 \langle v, \mu_C - \mu_g \rangle^t}{2}$. $\qquad \square$

Unfortunately, even the inliers might not form a clique in $G$. However, the guarantee that $|T' \cap T_g| \geq |T_g|/2$ implies that the inliers share many neighbors in the graph $G$. Thus we look at the overlap graph defined below, in the hope that this graph will be more dense.

**Definition 3.4** (Overlap Graph). *Let $G = (V, E)$ be a graph and $\gamma > 0$. The overlap graph $R_\gamma(G)$ is defined to be the graph with the vertex set $V$ where each $(x, y)$ is an edge in the graph iff $|N_G(x) \cap N_G(y)| \geq \gamma|V|$, where $N_G(x)$ denotes the neighborhood of the vertex $x$ in $G$.*

The following result shows that if $G$ has bounded moments, then so does $R_\gamma(G)$. While this is not sufficient by itself, we will subsequently modify $R_\gamma(G)$ to ensure that we end up with a graph having bounded moments, as well as an identifiable clique.

**Lemma 3.5** (If $G$ has bounded moments, then $R_\gamma(G)$ has bounded moments). *Let $S$ be a set of points and $G$ be a graph with $(M, t)$-bounded moments in $k$-sparse directions. Then for $\gamma > 0$, $R_\gamma(G)$ has $(2 \cdot 2^t M/\gamma, t)$-bounded moments in $k$-sparse directions.*

*Proof.* For any $x, y$ in $R_\gamma(G)$, the triangle inequality implies $\langle v, (x-a)-(a-y)\rangle^t \leq 2^t[\langle v, x-a\rangle^t + \langle v, y-a\rangle^t]$. By taking a sum over all $a$ in $N_G(x) \cap N_G(y)$, we have

$$\langle v, x-y\rangle^t = \sum_{a \in N_G(x) \cap N_G(y)} \frac{\langle v, (x-a)-(a-y)\rangle^t}{|N_G(x) \cap N_G(y)|} \leq (2^t/\gamma|S|) \sum_{a \in N_G(x) \cap N_G(y)} [\langle v, x-a\rangle^t + \langle v, y-a\rangle^t].$$

Denote $\Gamma(\alpha, x) := \{y : \text{neighbor of } x \text{ in } R_\gamma(G) \text{ and neighbor of } a \text{ in } G\}$. Summing over all the edges $(x, y)$ in $R_\gamma(G)$, and since $t$ is even and $G$ has $(M, t)$ bounded moments in $k$-sparse directions, we see: $\sum_{(x,y) \in R_\gamma(G)} \langle v, x-y\rangle^t \leq \frac{2^t}{\gamma|S|} \sum_{(x,y) \in R_\gamma(G)} \sum_{a \in N_G(x) \cap N_G(y)} [\langle v, x-a\rangle^t + \langle v, y-a\rangle^t] = \frac{2 \cdot 2^t}{\gamma|S|} \sum_{(a,x) \in E} \sum_{y \in \Gamma(\alpha, x)} \langle v, x-a\rangle^t \leq \frac{2 \cdot 2^t \cdot |S|}{\gamma|S|} \sum_{(a,x) \in E} \langle v, a-x\rangle^t \leq \frac{2 \cdot 2^t M |S|^2}{\gamma}$. $\square$

While $R_\gamma(G)$ may not have any cliques either, it is guaranteed to have fairly dense subgraphs. We will subsequently prune out points so that the resulting final graph will have a clique.

**Lemma 3.6** ($R_\gamma(G)$ has dense subgraphs). *Let $G = (V, E)$ be a graph and $\gamma > 0$. If $x$ is a randomly selected vertex of $G$, then the expected number of pairs $y, z \in N_G(x)$ so that $y$ and $z$ are not neighbors in $R_\gamma(G)$ is at most $\gamma|V|^2$.*

*Proof.* The expectation in question is $1/|V|$ times the number of triples $x, y, z \in V$ so that $y$ and $z$ are not neighbors in $R_\gamma(G)$, but are both neighbors of $x$ in $G$. By the definition of $R_\gamma(G)$, if $y$ and $z$ are not neighbors in $R_\gamma(G)$, they have at most $\gamma|V|$ common neighbors in $G$. Thus, the number of such triples is at most $\gamma|V|^3$, so the expectation in question is at most $\gamma|V|^2$. $\square$

As outlined above, the inliers in $R_\gamma(G)$ form a dense subgraph. The next procedure (PRUNING in Algorithm 2) prunes out points from a dense subgraph (inliers in $R_\gamma(G)$ for us) to find a clique.

---

**Algorithm 2** Algorithms for clique creation and rounding

1: **function** PRUNING($G = (V, E), W \subset V$)
2:      Let $W' = W$
3:      **while** $\exists x \in W'$ that is not connected to at least $2|W|/3$ vertices in $W'$: Remove $x$ from $W'$
4:      **return** $W'$
5: **end function**
6: **function** ROUNDING($S, G = (V, E)$)
7:      Let $\delta = \alpha^3/4608$.
8:      Choose $x \in S$ uniformly at random, let $W = N_G(x)$, and let $G' = R_\delta(G)$
9:      **if** the number of pairs of points in $W$ that are not connected in $G'$ is more than $(8\delta/\alpha)|V|^2$
       or if $|W| \leq (\alpha/4)|V|$ **return** FAIL
10:      **else** Run PRUNING on $G'$ and $W$ to obtain $W'$
11:      **return** $\mathbf{E}_{X \sim W'}[X]$.
12: **end function**

---

**Lemma 3.7** (Dense subgraphs can be pruned to obtain a clique). *Let $G = (V, E)$ be a graph and let $W \subset V$ be a set of vertices with $|W| = \beta|V|$ and all but $\gamma|V|^2$ pairs of vertices in $W$ are connected in $G$, for $\beta, \gamma > 0$ with $\gamma \leq \beta^2/36$. There exists an algorithm (PRUNING in Algorithm 2) that given $G, W, \beta, \gamma$ runs in polynomial time and returns a $W' \subset W$ so that $|W'| \geq |W| - (6\gamma/\beta)|V|$ and so that $|W'|$ is a clique in $R_{\beta/3}(G)$.*

*Proof.* In Line 3 of the Algorithm, the point $x$ which is removed satisfies $|N_G(x) \cap W'| < 2/3|W|$. If we also have that $|W'| \geq 5|W|/6$ (something that we will verify later), the removal of $x$ decreases the number of pairs of unconnected elements in $W'$ by $|W'| - |N_G(x) \cap W'| \geq |W'| - (2/3)|W| \geq |W|/6 = (\beta/6)|V|$. This can happen at most $(6\gamma/\beta)|V|$ times before we run out of unconnected pairs of elements in $W'$, thus $|W'| \geq |W| - (6\gamma/\beta)|V|$ upon termination. Also, since it holds $(6\gamma/\beta)|V| \leq (\beta/6)|V| = |W|/6$, we indeed have $|W'| \geq 5|W|/6$ as claimed at the start. Now note that each element of $W'$ is connected to at least $2|W|/3$ other elements of $W'$ in $G$. Thus any pair of elements of $W'$ have at least $|W|/3$ common neighbors, and thus are adjacent in $R_{\beta/3}(G)$. $\square$

We are finally ready to prove our main algorithmic result on rounding. The basic idea is that most of the inliers in $G$ (which are at least $\alpha$-fraction of vertices) are connected to many other inliers, and

thus if we start with an inlier, its neighborhood will also contain many inliers and will be dense in the overlap graph $G'$ (Lemma 3.6). Thus, we can apply the pruning of Lemma 3.7 to obtain a large clique in the overlap graph of $G'$, which also has bounded moments by two applications of Lemma 3.5.

**Theorem 3.8** (Rounding). *Let $S \subset \mathbb{R}^n$ and let $G = (V, E)$ be a graph with $V = S$ and $(M, t)$ bounded moments in $k$-sparse directions. Suppose there is a subset $S_g \subset S$ with $|S_g| \geq \alpha|S|$ and at least half of the pairs of points in $S_g$ are connected by an edge in $G$. Suppose that the $S_g$ has mean $\mu_g$ and $t$-th moment bounded by $M$ in $k$ sparse directions. Then, there exists a randomized algorithm that given $G, S$ and $\alpha$ runs in polynomial time and returns a $\widehat{\mu} \in \mathbb{R}^n$ such that with probability $\Omega(\alpha)$, for all $k$-sparse unit vectors $v$, $\langle v, \widehat{\mu} - \mu_g \rangle^t = O(10^t M \alpha^{-6})$*

*Proof.* The algorithm we consider is ROUNDING (Algorithm 2). Let $x$ and $G'$ be as in Line 8. We will claim that algorithm ROUNDING succeeds as long as the following hold: (i) $x \in S_g$, (ii) $x$ has at least $S_g/4$ neighbors in $S_g$ in $G'$, and (iii) the number of pairs of neighbors of $x$ that are not neighbors in $G'$ is at most $(8\delta/\alpha)|V|^2$. First, we show that these conditions hold with probability $\Omega(\alpha)$. The first condition holds with probability at least $\alpha$ over the choice of $x$. Conditioned on $x \in S_g$, the expected number of non-neighbors that $x$ has in $S_g$ is at most $|S_g|/2$. Thus, the probability that it has more than $3|S_g|/4$ non-neighbors is at most $2/3$ by Markov's inequality. Thus, the first two conditions both hold with probability at least $\alpha/3$. Finally, the expected number of pairs of neighbors of $x$ that are non-neighbors in $G'$ is at most $\delta|V|^2$ by Lemma 3.6. Thus, by Markov's inequality, there will be more than $(8\delta/\alpha)|V|^2$ such non-connected neighbors with probability at most $\alpha/8$. Combining with the above, all three conditions hold with probability at least $\alpha/24$.

Given these assumptions, we note that $|W| \geq |S_g|/4 \geq (\alpha/4)|V|$, and at most $(8\delta/\alpha)|V|^2$ of pairs in $W$ are not connected in $G'$. This implies that we pass the condition in Line 9. We will now verify the conditions in Lemma 3.7. Since $\beta := |W|/|V| \geq \alpha/4$ and $\gamma$, the number of pairs of vertices in $W$ that are not connected in $G'$, is at most $8\delta/\alpha = \alpha^2/576$, we have $\gamma \leq \beta^2/36$, satisfying the assumptions of Lemma 3.7. Thus the returned $W'$ is a clique in $R_{\beta/3}(G')$ and satisfies

$$|W| - |W'| \leq (6\gamma/\beta)|V| \leq (48\delta/\alpha)/(\alpha/4)|V| \leq (\alpha/24)|V|.$$

This means that $|W' \cap S_g| \geq |S_g|/4 - (\alpha/24)|V| \geq |S_g|/6$. On the other hand, we know that $G$ has $(M, t)$ bounded moments in $k$-sparse directions. Lemma 3.5 implies that $G'$ has moments bounded by $O(2^t M/\alpha^3)$. Applying the lemma once more implies that $R_{\beta/3}(G')$ has moments bounded by $O(4^t M/(\alpha^3\beta)) = O(4^t M/\alpha^4)$. Since $W'$ is a clique in $R_{\beta/3}(G')$, we have by Lemma 3.3 that if $\tilde{\mu}$ is the sample mean of $S_g \cap W'$, then

$$\langle v, \widehat{\mu} - \tilde{\mu} \rangle^t \leq O(4^t \, M\alpha^{-6}) \quad \text{for all } k\text{-sparse unit vectors } v \ . \tag{2}$$

Since $|S_g \cap W'| \geq |S_g|/6$ and the $S_g$ has bounded $t$-th moment along $k$-sparse directions, we have that $\langle v, \tilde{\mu} - \mu_{\text{good}} \rangle^t \leq O(M)$ (see Lemma B.1 for a proof of this fact). Combining this with Equation (2) using triangle inequality completes the proof. $\square$

### 3.3 Proof Sketch of Theorem 1.2

We sketch the proof here, deferring the full proof to Appendix B.1. Let $S$ be the $(1 - \alpha)$-corrupted set of samples, and $S_g$ be the inliers. Let $T = \{x - y \mid x, y, \in S\}$ and $T_g$ be its part due to inliers. To every subset $T'$ of $T$, we can associate a graph $G_{T'}$ having vertices $S$ and edges between the pairs included in $T'$. Because of Lemma 2.4, $T_g$ has certifiably bounded moments in $k$-sparse directions. By Theorem 3.1, the filtering step will return a subset $T' \subset T$ that has sizable overlap with $T_g$ and its graph $G_{T'}$ has bounded moments in $k$-sparse directions. Finally, By Theorem 3.1, the rounding algorithm will return a $\hat{\mu}$ that is close to $\mu$ in all $k$-sparse directions. This $\hat{\mu}$ can be truncated to yield a vector close to $\mu$ in the standard $\ell_2$-norm (Fact A.2).

## 4 Acknowledgments

ID was supported by NSF Medium Award CCF-2107079, NSF Award CCF-1652862 (CAREER), a Sloan Research Fellowship, and a DARPA Learning with Less Labels (LwLL) grant. DK was supported by NSF Medium Award CCF-2107547, NSF Award CCF-1553288 (CAREER), a Sloan Research Fellowship, and a grant from CasperLabs. SK was supported by NSF under Grant #2127309 to the Computing Research Association for the CIFellows 2021 Project. AP was supported by NSF grants NSF Award CCF-1652862 (CAREER), DMS-1749857, and CCF-1841190. TP was supported by NSF Award CCF-1652862 (CAREER) and NSF Medium Award CCF-2107079.

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
