# Supplementary Material

## A Omitted Background

**Basic Notation.** We use $\mathbb{N}$ to denote natural numbers and $\mathbb{Z}_+$ to denote positive integers. For $n \in \mathbb{Z}_+$ we denote $[n] := \{1, \ldots, n\}$. We denote by $\mathbb{R}[x_1, \ldots, x_n]_{\leq d}$ the class of real-valued polynomials of degree at most $d$ in variables $x_1, \ldots, x_n$. We use $\text{poly}(\cdot)$ to indicate a quantity that is polynomial in its arguments. For an ordered set of variables $Q = \{x_1, \ldots, x_n\}$, we will denote $p(Q)$ to mean $p(x_1, \ldots, x_n)$. Throughout the paper, we will typically use the letter $n$ for the dimension, $m$ for the number of samples, $d$ for the degrees of the SoS proofs, and $t$ for the number of bounded moments.

**Linear Algebra Notation.** We use $I_n$ to denote the $n \times n$ identity matrix. We will drop the subscript when it is clear from the context. We typically use small case letters for deterministic vectors and scalars. We will specify the dimensionality unless it is clear from the context. We denote by $e_1, \ldots, e_n$ the vectors of the standard orthonormal basis, i.e., the $j$-th coordinate of $e_i$ is equal to $\mathbf{1}_{\{i=j\}}$, for $i, j \in [n]$. For a vector $v$, we let $\|v\|_2$ denote its $\ell_2$-norm. We call a vector $k$-sparse if it has at most $k$ non-zero coordinates. We use $\langle v, u \rangle$ for the inner product of the vectors $u, v$. We will use $\cdot^{\otimes s}$ to denote the standard Kronecker product.

**Probability Notation.** We use capital letters for random variables. For a random variable $X$, we use $\mathbf{E}[X]$ for its expectation. We use $\mathcal{N}(\mu, \Sigma)$ to denote the Gaussian distribution with mean $\mu$ and covariance matrix $\Sigma$. We let $\phi$ denote the pdf of the one-dimensional standard Gaussian. When $D$ is a distribution, we use $X \sim D$ to denote that the random variable $X$ is distributed according to $D$. When $S$ is a set, we let $\mathbf{E}_{X \sim S}[\cdot]$ denote the expectation under the uniform distribution over $S$. For any sequence $a_1, \ldots, a_m \in \mathbb{R}^n$, we will also use $\mathbf{E}_{i \sim [m]}[a_i]$ to denote $\frac{1}{m} \sum_{i \in [m]} a_i$. For a real-valued random variable $X$ and $p \geq 1$, we use $\|X\|_{L_p}$ to denotes its $L_p$ norm, i.e., $\|X\|_{L_p} := (\mathbf{E}[|X|^p])^{1/p}$.

**Definition A.1** ($(2, k)$-norm). *We define the $(2, k)$-norm of a vector $x$, denoted as $\|x\|_{2,k}$, to be the maximum correlation with any $k$-sparse unit vector, i.e., $\|x\|_{2,k} := \max_{\|v\|_2 = 1, v:k-\text{sparse}} \langle v, x \rangle$.*

The following standard fact translates bounds from the $(2, k)$-norm to the usual $\ell_2$-norm when the underlying mean $\mu$ is $k$-sparse (see, e.g., [DKK$^+$22a] for a proof):

**Fact A.2.** *Let $h_k : \mathbb{R}^n \to \mathbb{R}^n$ denote the function where $h_k(x)$ is defined to truncate $x$ to its $k$ largest coordinates in magnitude and zero out the rest. For all $\mu \in \mathbb{R}^n$ that are $k$-sparse, we have that $\|h_k(x) - \mu\|_2 \leq 3\|x - \mu\|_{2,k}$.*

### A.1 Additional SoS Preliminaries

**Definition A.3** (Symbolic polynomial). *A degree-$d$ symbolic polynomial $p$ is a collection of indeterminates $\widehat{p}(\alpha)$, one for each multiset $\alpha \subseteq [n]$ of size at most $d$. We think of it as representing a polynomial $p : \mathbb{R}^n \to \mathbb{R}$ whose coefficients are themselves indeterminates via $p(x) = \sum_{\alpha \subseteq [n], |\alpha| \leq t} \widehat{p}(\alpha) x^\alpha$.*

**Theorem A.4** (The SoS Algorithm [Sho87, Las01, Nes00, Bom98]). *For any $n, r, \ell \in \mathbb{Z}^+$, and a set of $r$ polynomial equalities $\mathcal{P} = \{p_1(x_1, \ldots, x_n) = 0, \ldots, p_r(x_1, \ldots, x_n) = 0\}$, the following set has an $(rn)^{O(\ell)}$-time weak separation oracle (in the sense of [GLS81]):*

$$\{q(x_1, \ldots, x_n) : \mathcal{P} \left|\frac{x_1, \ldots, x_n}{\ell}\right. q(x_1, \ldots, x_n) \geq 0\}$$

It is standard fact that many inequalities like Cauchy-Schwartz and the triangle inequality have a Sum of Squares version. We will use these extensively.

**Fact A.5** (SoS Cauchy-Schwartz and Hölder (see, e.g., [Hop18])). *Let $f_1, g_1, \ldots, f_n, g_n$ be indeterminates. Then,*

$$\left|\frac{f_1, \ldots, f_n, g_1, \ldots, g_n}{2}\right. \left\{ \left(\frac{1}{n} \sum_{i=1}^n f_i g_i\right)^2 \leq \left(\frac{1}{n} \sum_{i=1}^n f_i^2\right) \left(\frac{1}{n} \sum_{i=1}^n g_i^2\right) \right\}.$$

**Fact A.6** (SoS Triangle Inequality). *If $k$ is a power of two, $\left|\frac{a_1, a_2, \ldots, a_n}{k}\right. \left\{ \left(\sum_i a_i\right)^k \leq n^k \left(\sum_i a_i^k\right) \right\}$.*

## A.2 Martingales

**Definition A.7** (Submartingale)**.** *A submartingale is an integer-time stochastic process* $\{X_i \mid i \in \mathbb{Z}_+\}$ *that satisfies the following:*

1. $\mathbf{E}[|X_i|] < \infty$.

2. $\mathbf{E}[X_i | X_{i-1}, X_{i-2}, \ldots, X_1] \geq X_{i-1}$.

**Fact A.8** (Optimal Stopping Theorem)**.** *Let* $X_1, X_2, \ldots$ *be a sub-martingale and* $T$ *be a finite stopping time. Then,* $\mathbf{E}[X_T] \geq \mathbf{E}[X_1]$.

**Proposition A.9.** *Let* $X_1, X_2, \ldots$ *be a sub-martingale for which* $0 \leq X_i \leq 1$ *almost surely and fix an integer* $n < \infty$. *Then, for any* $t \in (0, 1)$*, we have that*

$$\Pr\left[\min_{1 \leq i \leq n} X_i \geq t\right] \geq \frac{\mathbf{E}[X_1] - t}{1 - t} \ .$$

*Proof.* Let the random variable $T$ defined as the minimum between $n$ and $\arg\min_i\{X_i < t\}$. Then $T$ is a stopping time and it is finite. By the optimal stopping theorem, $\mathbf{E}[X_T] \geq \mathbf{E}[X_1]$. Also, for every random variable $Y \in [0, 1]$ and $t \in (0, 1)$, Markov's inequality implies that $\Pr[Y \geq t] \geq (\mathbf{E}[Y] - t)/(1 - t)$. Using the two, we have that

$$\Pr\left[\min_{1 \leq i \leq n} X_i < t\right] \leq \Pr\left[\mathbf{E}[X_T] < t\right] < 1 - \frac{\mathbf{E}[X_1] - t}{1 - t} \ .$$

$\square$

The following lemma has its proof in [DKK+22a]. Using this, it is possible to show that $O(t^2 \log(n))$ moments being bounded is sufficient to show concentration of the $t$-th tensors in $\ell_\infty$ norm.

**Lemma A.10.** *Let* $D$ *be a distribution over* $\mathbb{R}^n$ *with mean* $\mu$*. Suppose that for all* $s \in [1, \infty)$*,* $D$ *has its* $s^{th}$ *moment bounded by* $(f(s))^s$ *for some non-decreasing function* $f : [1, \infty) \to \mathbb{R}_+$*, in the direction* $e_j$*, i.e., suppose that for all* $j \in [n]$ *and* $X \sim D$:

$$\|\langle e_j, X - \mu \rangle\|_{L_s} \leq f(s).$$

*Let* $X_1, \ldots, X_m$ *be* $m$ *i.i.d. samples from* $D$ *and define* $\overline{\mu} := \sum_{i=1}^m X_i$*. The following are true:*

1. *If* $m \geq \max\left(\frac{1}{\delta^2}, 1\right) C\left(t \log(n/\gamma)\right) \left(2f(t^2 \log(n/\gamma))\right)^{2t} \max\left(1, \frac{1}{f(t)^{2t}}\right)$*, then with probability* $1 - \gamma$*, we have that*

$$\left\|\mathop{\mathbf{E}}_{i \sim [m]}[(X_i - \overline{\mu})^{\otimes t}] - \mathop{\mathbf{E}}_{X \sim D}[(X - \mu)^{\otimes t}]\right\|_\infty \leq \delta \ .$$

2. *If* $m > C(k/\delta^2) \log(n/\gamma)(f(\log(n/\gamma)))^2$*, then with probability* $1 - \gamma$*, it holds*

$$\left\|\mathop{\mathbf{E}}_{X \sim S}[X] - \mu\right\|_{2,k} \leq \delta.$$

# B  Omitted Proofs from Section 3

**Lemma B.1.** *Let* $t \in \mathbb{Z}_+$ *even. Let* $\mathcal{U}$ *be a set of unit vectors in* $\mathbb{R}^n$ *and* $S$ *be a set with* $t$*-th central moment bounded by* $M$ *in the directions of* $\mathcal{U}$*, i.e.,* $\mathbf{E}_{X \sim S}[\langle v, X - \mathbf{E}_{X \sim S}[X] \rangle^t] \leq M$ *for all* $v \in \mathcal{U}$*. Then for all* $T \subset S$ *with* $|T| \geq \alpha|S|$*, if we denote by* $\mu_S$ *and* $\mu_T$ *the means of* $S$ *and* $T$ *respectively, we have that*

$$\langle v, \mu_S - \mu_T \rangle^t \leq \frac{M}{\alpha} \ ,$$

*for all* $v \in \mathcal{U}$*.*

*Proof.* Let $\mu_S := \mathbf{E}_{X \sim S}[X]$ and $\mu_T := \mathbf{E}_{X \sim T}[X]$. We have that

$$M \geq \mathop{\mathbf{E}}_{X \sim S}[\langle v, X - \mu_S \rangle^t] \geq \alpha \mathop{\mathbf{E}}_{X \sim T}[\langle v, X - \mu_S \rangle^t] \geq \alpha \langle v, \mu_T - \mu_S \rangle^t \ ,$$

where the last inequality uses Jensen's inequality. $\square$

## B.1 Proof of Theorem 1.2

We restate the main theorem below.

**Theorem 1.2** (List-Decodable Sparse Mean Estimation). *Let $t$ be an integer power of two. Let $D$ be a distribution over $\mathbb{R}^n$ with $k$-sparse mean $\mu$. Suppose that $D$ has $t$-th moments $d$-certifiably bounded in $k$-sparse directions by $M$ for some $d = O(t)$ (cf. Definition 2.3) and subexponential tails in the standard basis directions. There is an algorithm which, given $\alpha$, $M$, $t$, $k$, and a $(1-\alpha)$-corrupted set of $m = (tk \log n)^{O(t)} \max(1, M^{-2})/\alpha$ samples from $D$, runs in time $\mathrm{poly}(mn^t)$ and returns a vector $\hat{\mu} \in \mathbb{R}^n$ such that with probability $\Omega(\alpha)$ it is the case that $\|\hat{\mu} - \mu\|_2 = O_t(M^{1/t}/\alpha^{O(1)/t})$.*

---

**Algorithm 3** Algorithm for list-decodable sparse mean estimation.

---

1: **function** LDSPARSE-MEAN($S = \{x_1, \ldots, x_m\}, \alpha, M, t, k$)
2:     Let $C \in \mathbb{Z}_+$ be a large enough constant ($C > 5$ suffices).
3:     Form the set $T = \{x - y \mid x, y \in S\}$
4:     $T' \leftarrow$ LDMEAN-SoS-FILTER($T, t, Ct, M$)
5:     Let $G = (V, E)$ with $V = S$ and $E = \{(x, y) : x - y$ or $y - x$ belongs in $T'\}$.
6:     $\hat{\mu} \leftarrow$ ROUNDING($S, G$).
7:     Let $h_k : \mathbb{R}^n \to \mathbb{R}^n$ denote the function where $h_k(x)$ is defined to truncate $x$ to its $k$ largest coordinates in magnitude and zero out the rest.
8:     **return** $h_k(\hat{\mu})$.
9: **end function**

---

*Proof.* Let $S$ be the $(1 - \alpha)$-corrupted set of samples given as input to the algorithm and $S_{\mathrm{g}}$ be the subset of $S$ corresponding to the inliers. Given $S$, construct the set of differences $T := \{x - y \mid x, y \in S\}$. Also, denote by $T_{\mathrm{g}}$ the same set of corresponding to the inliers.

For the inliers, the number of samples is large enough so that with constant probability the conclusion of Lemma 2.4 holds. We thus condition on this event for the rest of the proof. Its first part states that $S_{\mathrm{g}}$ has $(M', t, d)$-certifiable bounded moments in $k$-sparse directions, where $M' = 8M$ and $d = O(t)$. By SoS triangle inequality, $T_{\mathrm{g}}$ (Fact A.6) has $(2^t M', t, O(d))$ bounded central moments in $k$-sparse directions.

Now, Theorem 3.1 identifies a subset $T' \subset T$ such that with probability at least $2/3$:

1. For all $k$-sparse unit vectors $v$ it holds $\sum_{x \in T'} \langle v, x \rangle^t \leq 6 \cdot 2^t M' |T|$.

2. We have $|T' \cap T_{\mathrm{g}}| \geq |T_{\mathrm{g}}|/2$.

Construct the graph $G = (V, E)$ with vertex set $V = S$ and edges $(x, y)$ for every pair of $x, y$ that $x - y$ or $y - x$ is in $T'$. By Item 1 above, $G$ has $(6 \cdot 2^t M', t)$ bounded moments in $k$-sparse directions. By Item 2, at least half of the pairs of points in $S_{\mathrm{g}}$ are connected by an edge in $G$. Moreover, $S_{\mathrm{g}}$ has $(M', t, d)$-certifiable bounded moments in $k$-sparse directions for $d \geq t$. These are the conditions of Theorem 3.8, thus an application of this to the graph $G$ yields that for every $k$-sparse unit vector $v$ we have that

$$\langle v, \hat{\mu} - \mu_{\mathrm{g}} \rangle^t = O(10^t M' \alpha^{-6}).$$

Also, by the second part of the conclusion of Lemma 2.4, we have that $\langle v, \mu - \mu_{\mathrm{g}} \rangle^t \leq M' \alpha^{-6}$ for every $k$-sparse unit vector $v$. Using the triangle inequality, we have that $\langle v, \mu - \hat{\mu} \rangle^t \leq O(20^t M' \alpha^{-6})$ for every $k$-sparse unit vector $v$. Then, Fact A.2 provides a way to truncate the vector $\hat{\mu}$ so that the result $h_k(\hat{\mu})$ satisfies $\|h_k(\hat{\mu}) - \mu\|_2^t = O(M' \alpha^{-6}) = O(M \alpha^{-6})$. Raising both sides to the power $1/t$ gives the desired claim.

$\qquad\square$

## C Information-Computation Tradeoffs

In this section we present evidence of an information-computation gap for our problem, that is, we provide evidence that computationally efficient list-decoding algorithms for sparse mean estimation

of distributions with bounded $t$-th moments up to error $O(\alpha^{-c/t})$ might inherently need more samples than what is needed to get the same error by computationally inefficient algorithms. More specifically, we give statistical query and low-degree polynomial testing lower bounds for list-decodable sparse mean estimation, which indicate that the factor $k^{O(t)}$ appearing in the sample complexity of our algorithm from the previous sections might be necessary for computational efficiency. This is to be compared with the fact that, for distributions with bounded $t$-th moments, it is information-theoretically possible to identify a list of $O(1/\alpha)$ candidate vectors, containing at least one that is within euclidean distance $O(\alpha^{-1/t})$ using $O(k \log n)/\alpha^3$ samples.[3]

In the statistical query model, algorithms are allowed only to perform queries of the following kind instead of drawing samples.

**Definition C.1** (STAT Oracle). *Let $D$ be a distribution on $\mathbb{R}^n$. A statistical query is a bounded function $f : \mathbb{R}^n \to [-1, 1]$. For $\tau > 0$, the $\mathrm{STAT}(\tau)$ oracle responds to the query $f$ with a value $v$ such that $|v - \mathbf{E}_{X \sim D}[f(X)]| \leq \tau$. We call $\tau$ the tolerance of the statistical query.*

The results of this section follow by simple modification of previous work of [DKS18a]. We thus do not include self contained proofs here but mention only the key differences. We start by formally defining the problem of list-decodable sparse mean estimation. In fact our lower bound would hold against even a weaker noise model, where the noise is i.i.d. from an arbitrary distribution.

**Problem C.2** (List-Decodable Sparse Mean Estimation). Fix $\rho > 0$ and $\alpha \in (0, 1/2)$. Given access to the mixture distribution $\alpha \mathcal{N}(\rho v, I_n) + (1 - \alpha)B$, for some (unknown) $k$-sparse unit vector $v$ in $\mathbb{R}^n$ and some (unknown and arbitrary) distribution $B$, the goal is to find a list of vectors $\mathcal{L}$ with the guarantee that there exists a $u \in \mathcal{L}$ such that $\|u - \mathbf{E}_{X \sim D}[X]\|_2 < \rho/4$.

The lower bounds of this section will in fact be about the more basic hypothesis testing version of the problem.

**Problem C.3** (Hypothesis Testing of List-Decodable Sparse Means). Fix $\rho > 0$. We define the following hypothesis testing problem:

- $H_0$: The underlying distribution is $\mathcal{N}(0, I_n)$.

- $H_1$: The underlying distribution is $\alpha \mathcal{N}(\rho v, I_n) + (1 - \alpha)B$, for some unknown $k$-sparse unit vector $v$ in $\mathbb{R}^n$ and some unknown distribution $B$.

It is known that the two problems are related by the following reduction. The resulting algorithm is known to be implementable in both the statistical query and the low-degree polynomials model.

**Fact C.4** ([DKP+21]). *Fix $\rho > 0$ and the dimension $n \in \mathbb{Z}_+$. Denote by $\mathcal{A}$ an algorithm that, whenever given some access to the distribution $\alpha \mathcal{N}(\rho v, I_n) + (1 - \alpha)B$ with unknown $B, v$, it returns a list $\mathcal{L}$ of candidate vectors such that there exists $u \in \mathcal{L}$ with $\|u - \rho v\|_2 \leq \rho/4$. Then, there exists a procedure that calls $\mathcal{A}$ twice and solves the hypothesis testing Problem C.3 with probability at least $1 - |\mathcal{L}|^2/n$. The running time of this reduction is quadratic in $|\mathcal{L}|n$.*

*Proof.* We follow the same proof strategy as [DKP+21, Lemma 5.9] with a crucial modification: instead of using the random rotation matrix $A$ in [DKP+21, Algorithm 1], we use a special kind of rotation matrix that can only shuffle the coordinates and flip the signs (see Fact C.5 below). This modification is needed because the latter family of rotation matrices preserve the sparsity of vectors. We obtain the desired conclusion by following the same proof as [DKP+21] but replacing [DKP+21, Lemma 5.10] with the following claim :

**Fact C.5.** *Let $\sigma_1, \ldots, \sigma_n$ be $n$ independent Rademacher random variables. Let $A$ be an $n \times n$ independent permutation matrix generated uniformly at random. Let $A'$ be the matrix generated by multipliying the $i$-th row of $A$ by $\sigma_i$ for each $i \in [n]$, i.e, $A'_{i,j} = \sigma_i A_{i,j}$. For any fixed vectors $u$ and $v$, let $Z := \langle u, A'v \rangle$. Then $\mathbf{E}[Z] = 0$ and the variance of $Z$ is $\|u\|_2^2 \|v\|_2^2/n$.*

*Proof.* Let $Z = \langle u, A'v \rangle$ and observe that $Z = \sum_{i,j} \sigma_i u_i A_{i,j} v_j$. Since $\sigma_i$'s are zero mean, we have that $\mathbf{E}[Z] = 0$. To calculate the variance, we use the following facts: (i) For any $i \in [n]$, we have that

---

[3]This result is shown for the dense case in [DKS18b]; the adaptation to the sparse case follows immediately by taking a union bound over the $\binom{n}{k}$ coordinates before applying the VC concentration inequality.

$A_{i,j}A_{i,\ell} = 0$ almost surely if $j \neq l$, and (ii) $\mathbf{E}[A_{i,j}^2] = \mathbf{E}[A_{i,j}] = 1/n$. Using these, we obtain the following expression for the variance of $Z$.

$$\mathbf{Var}(Z) = \mathbf{E}\left[\left(\sum_{i,j}\sigma_i u_i A_{i,j} v_j\right)^2\right] = \mathbf{E}\left[\sum_{i,j,k,\ell}\sigma_i\sigma_k u_i v_j u_k v_\ell A_{i,j}A_{k,\ell}\right]$$

$$= \mathbf{E}\left[\sum_{i,j,\ell}\sigma_i^2 u_i^2 v_j v_\ell A_{i,j}A_{i,\ell}\right] = \mathbf{E}\left[\sum_{i,j}u_i^2 v_j^2 A_{i,j}^2\right] = \sum_{i,j}(u_i^2 v_j^2)/n = \|u\|_2^2\|v\|_2^2/n.$$

where the third equation is because $\mathbf{E}[\sigma_i\sigma_j] = 0$ if $i \neq j$ and the next one because of $A_{i,j}A_{i,\ell} = 0$ if $j \neq \ell$. $\qquad\square$

Since the variance is bounded, we can apply the Chebyshev's inequality to get an upper bound on the failure probability of the reduction. $\qquad\square$

We now state the SQ lower bound and sketch its proof.

**Theorem C.6** (Statistical Query Lower Bound). *Let $k, n, t \in \mathbb{Z}_+$ with $k \leq \sqrt{n}$, and $c > 0$ be a small enough constant. Let $\mathcal{A}$ be an SQ algorithm that solves the hypothesis testing Problem C.3 with $\rho = c(t\alpha)^{-1/t}$. Then, $\mathcal{A}$ does one of the following:*

- *it uses at least one query with tolerance $O\left(2^{t/2}k^{-(t+1)/4}\exp\left(O((t\alpha)^{-2/t})\right)\right)$ or*

- *it makes $\Omega\left(n^{\sqrt{k}/16}k^{-(t+1)/2}\right)$ many queries.*

*Proof.* Let $A$ be the one-dimensional distribution of [DKS18b, Lemma 5.5], which satisfies the following properties: (i) $A = \alpha\mathcal{N}(\rho, 1) + (1-\alpha)E$ for some distribrution $E$, (ii) $A$ matches the first $t$ moments with $\mathcal{N}(0,1)$, and $\chi^2(A, \mathcal{N}(0,1)) := \int_{-\infty}^{+\infty}(A(x) - \phi(x))^2/\phi(x)\mathrm{d}x = \exp(O(t\alpha)^{-2/t})$, where $\phi(x)$ denotes the pdf of $\mathcal{N}(0,1)$. Then, the result follows from [DKK+22a, Corollary 6.7]. $\qquad\square$

Instead of using a reduction to the hypothesis testing problem, one can also obtain the exact same lower bound directly against the list-decodable mean estimation algorithms (i.e., search version of the problem as opposed to the decision version of the problem) by using the framework of [DKS17, DKS18c]; see, for example, Theorem 1.3. The reduction to the hypothesis testing problem outlined here is provided for two reasons: (i) it is conceptually insightful, and (ii) it allows us to show lower bounds against low-degree polynomial tests (see the remark below).

**Remark C.7.** By using the equivalence between SQ and low-degree polynomials [BBH+21], Theorem C.6 also implies qualitatively similar lower bound holds against low-degree polynomial tests. Specifically, the relevant statement for our case is obtained by using [DKK+22a, Theorem 6.23] with $m = t$, with the following interpretation: Unless the number of samples used is greater than $k^{(1-c)(t+1)}/(2^{t+1}\chi^2(A, \mathcal{N}(0, I_n)))$, any polynomial of degree roughly up to $k^c \log n$ fails to provide a good test for the hypothesis testing problem of Problem C.3. We refer to [BBH+21] for the formal definitions that quantify the notion of goodness of polynomial tests.

## D Adaptations of Prior Work to the Sparse Setting

Here we comment on prior SoS-based techniques for list-decodable (dense) mean estimation [KS17, RY20a]. While these prior techniques can be plausibly adapted to the sparse setting with some effort to match our guarantees qualitatively, this would result in significantly more complicated algorithms. We briefly point out some difficulties below.

First, to ensure that the algorithms identify subsets of the samples that satisfy certifiably bounded moments in all $k$-sparse directions requires additional variables and constraints to the algorithms. Additionally, one would need to replace the bounded moments in all directions condition by the corresponding condition for the sparse case, and ensure that all the proof steps can be modified to

rely only on the latter – this would result in minor modifications of the original algorithms, such as thresholding of candidate solution vectors.

Second, at the end of this process, while the algorithm might qualitatively match the error guarantee that we achieve, the runtime would continue to be $(mn)^{O(t)}$ for [KS17] or $(1/\alpha)^{\mathrm{polylog}(1/\alpha)} n^{O(\max\{1/\alpha^4, t\})}$ for [RY20a] – both of these are qualitatively worse than the runtime we achieve when $\alpha$ is sufficiently small. To obtain improved runtime using these prior techniques, one would require additional ideas, e.g., from [ST21], to be adapted to this setting, overall resulting in a far more complicated algorithm.

On the other hand, it seems difficult to adapt the multi-filtering technique from [DKS18a] to the setting we consider without any introduction of an SoS component. We remind the reader that the [DKS18a] algorithm depends critically on knowing the higher moments of the good points *exactly*, and does not generalize to less specific distribution families.