# OpenReview forum: "List-Decodable Sparse Mean Estimation via Difference-of-Pairs Filtering"
_NeurIPS.cc/2022/Conference — NeurIPS 2022 Accept_

### Official Review · Reviewer_Ymm8 · 2022-07-06

**Rating:** 8
**Confidence:** 4
**Soundness:** 4 excellent
**Presentation:** 4 excellent
**Contribution:** 3 good

**Summary:**

The field of algorithmic robust statistics is concerned with the development of efficient algorithms for statistical estimation problems in settings where the observed data is extremely (often adversarially) noisy. For the canonical problem of mean estimation, a single grossly corrupted data point can completely invalidate the performance of the sample average as a natural estimate of the population mean. In settings usually considered in this domain, one assumes that the algorithm observes $n$ data points generated in the following manner:

1. First $\alpha n$ ``good'' data points are generated from the true underlying distribution $D$.
2. An adversary then inspects the generated samples and adds an arbitrary set of $(1 - \alpha) n$ points to the dataset. Note that the algorithm is given no knowledge of where the corrupted data points are.

Restricting to the specific problem of mean estimation in high dimensions where the data set $X = \left\\{x_i\right\\}_{i = 1}^n \subset \mathbb{R}^d$, the goal is to recover the mean of the distribution $D$ generating the good data points. This task is made complicated by the fact that the algorithm does not actually know which points these are and natural approaches such as distance based thresholding yield sub-optimal results. In the standard setting when $\alpha \in (1/2, 1]$, approximate identification of $\mu$ is possible with error (in Euclidean norm) ranging between $\sqrt{1 - \alpha}$ to $\alpha$ and computationally and statistically efficient estimators have been designed. In the more challenging list decoding setting when $\alpha < 1/2$, even approximate identification is not possible and one instead returns a list of $1 / \alpha$ estimates one of which is guaranteed to be close to $\mu$. Note that a list of this size is necessary by considering the special case where the data is generated from a mixture of $1 / \alpha$ well behaved distributions. Efficient estimators have also been proposed in this setting with the guarantee that at least one of the elements in the list is close to $\mu$. The degree of closeness ranges between $1 / \sqrt{\alpha}$ when only second moment assumptions are placed on $D$ with improvements possible when stronger restrictions are placed on the distribution. Typically, these involve higher order moments of the distribution and the recovery error correspondingly improves to $(1 / \alpha)^{O (1 / t)}$ if $t$ moments are available.

This paper considers the list decodable setting in the sparse regime where $\mu$ is assumed to be $k$-sparse and $D$ is a distribution centered at $\mu$. In line with prior work on sparse estimation, the goal of the paper is to build an estimator whose sample complexity depends very mildly on the ambient dimension. All prior work incur sample complexity at least $d / \alpha$ which while optimal under no additional assumptions may be prohibitive when the ambient dimension is too large. The paper constructs a polynomial-time estimator which achieves sample complexity $\mathrm{poly} (k \log d)^{O(t)} / \alpha$ achieving recovery error of $(1 / \alpha)^{O(1 / t)}$ which is essentially optimal up to constants in the exponent when the distribution has $t$ certifiably good moments in sparse directions. Furthermore, the paper proposes a novel conceptually simpler approach to list-decodable estimation which avoids the more intricate filtering techniques used in prior work.

In the ``multi-filter'' framework for building estimators in the list decodable setting, one starts with a candidate set of good data points and infers one of the following:

- The dataset is well behaved (in terms of its moments) and the empirical mean is a good estimate
- Alternatively, the dataset is poorly behaved but a certificate of this fact (usually a direction along which moments are not well concentrated) can be used to refine (remove bad points from) the dataset. One then constructs a set of subsets of the original dataset which are better behaved.

At the conclusion of this process, the empirical means of all the refined datasets are returned as candidates in the returned list. Carrying out this approach requires careful design of the filtering and pruning mechanisms to avoid exponential growth in the number and sizes of subsets obtained in the leaves. The intricacies of this approach make reasoning and analysis of these algorithms difficult motivating simpler alternatives.

In the paper, the authors propose a novel approach where the start by constructing the difference of all pairs of of points in $X$ to construct a new $n^2$ sized dataset $T$. Crucially, all points $y = x_i - x_j$ when both $x_i$ and $x_j$ are drawn from $D$ have zero mean and have well-behaved moments around $0$. Furthermore, these points comprise a substantial $\alpha^2$ fraction of the dataset $T$. The authors then devise a one dimensional filter that given searches for points significantly increasing the sparse-directional moments of the set of datapoints $T$. Formally, one searches for a solution to the following set of polynomial constraints:
\begin{equation*}
    \\left\\{z_i^2 = z_i\\right\\}\_{i \\in [d]} \\cup \\left\\{v_i z_i = v_i\\right\\}\_{i \\in [d]} \cup \\left\\{\sum_{i \in [d]} z_i = k\\right\\} \cup \\left\\{\sum_{i \in [d]} v_i^2 = 1\\right\\} \cup \\left\\{\sum_{x \in T} \langle x, v \rangle^t \geq 6M|T| \\right\\}.
\end{equation*}
The first four encode a sparsity constraint in the direction $v$ while the last searches for a direction with large directional moment. Since, the above polynomial optimization is non-convex and in general NP-Hard, the paper instead solves a sum-of-squares relaxation. If no solution to the above set of equations is found, the process terminates with a well behaved set of points. Otherwise, the algorithm uses the solution to prune the points in $T$ to a smaller set with better moment behavior. A large good set exists because the $(\alpha n)^2$ sized subset consisting of pairs from the true distribution exists in $T$ and these points are very rarely removed. This step is conceptually similar to simple filtering strategies for the conceptually simpler iterative filtering setting where $\alpha > 1/2$.

Having pruned $T$ to a set with well-defined moments, the authors then design an intuitive procedure to identify a large subset of $X$ with substantial overlap with the good set and well-behaved moments. To do this, they note that $T$ induces a graph with vertices $X$ and pairs retained by the pruning process. The good set is expected to form an approximate clique within this graph. However, identifying a large clique is a hard problem in general. However, the authors observe that the graph has several nice properties. The first being that one can add edges between points $(x_i, x_j)$ whose neighborhoods have substantial overlap which also results in a larger corresponding difference set with well-behaved moments. The authors then show that if a random point from the set of good points is chosen and its neighborhood set is pruned based on the number of neighbors within the set, this yields a set of points with well behaved moments and large overlap with the set of good points. The mean of these points then yields a good approximation to the true mean $\mu$ from these two properties. Finally, selecting a point at random results in an point in the good set with probability at $\Omega (\alpha)$. Thus repeating this procedure $O(1 / \alpha)$ times results in a list of size $O(1 / \alpha)$ one of which is a good approximation to the true mean.

**** POST REBUTTAL UPDATE ****

I acknowledge the authors response and have decided to retain my current evaluation.

**Questions:**

One question I have for the authors is the implications of these results for list-decodable sparse mean estimation with fewer moments. For example, in the setting where the underlying distribution only has $2$ moments (bounded covariance), does this approach achieve the optimal recovery of $O(1 / \sqrt{\alpha})$ with say $\mathrm{poly} (k, \log d, 1/\alpha$ samples? This is in line with prior work on list decodable estimation which yield conceptually simpler algorithms in the bounded covariance setting albeit with worse recovery guarantees due to the milder moment requirements.


**Limitations:**

Yes

**Strengths And Weaknesses:**

The paper makes good progress on the important problem of building robust estimators in the sparse regime. Furthermore, the natural framework proposed in the paper is substantially simpler than prior approaches to list-decodable estimation. However, it is not clear from the current draft the implications of this framework for other settings where much fewer moments are available in the good planted distribution. In particular, it is not clear if the exponent in the recovery guarantees are optimal when one is concerned with the heavy tailed regime.

---

> ### Author Response · Authors · 2022-08-02
> **Authors' response to Reviewer Ymm8**
>
> We thank the reviewer for their effort and positive assessment of our work. Indeed, in addition to what the reviewer has mentioned, we point out the specific difficulty of extending the multifilter approach to exploit higher order moments in the sparse setting in lines 150-154 of our paper.  We answer their specific question below:
>
> + *“One question I have for the authors is the implications of these results for list-decodable sparse mean estimation with fewer moments”* $\qquad\qquad\qquad\qquad\qquad\qquad\qquad\qquad\qquad\qquad\qquad\qquad\qquad\qquad\qquad\qquad\qquad\qquad\qquad\qquad\qquad$
>
>
>    Thank you for the insightful question. While the heavy-tailed setting is beyond the scope of the present work (indeed, our algorithms crucially use higher moments to obtain their error guarantees), in the dense setting with heavy-tailed distributions, we can recover the mean up to an error of $\alpha^{-1/3}$ given $\text{poly}(d)$ samples, which although sub-optimal, is nonetheless non-trivial. $\qquad\qquad\qquad\qquad\qquad\qquad\qquad\qquad\qquad\qquad\qquad\qquad\qquad\qquad\qquad\qquad\qquad\qquad\qquad\qquad\qquad$
>
>   When we consider the sparse setting with heavy-tailed distributions, additional challenges arise because one cannot use spectral concentration bounds (as they require $\text{poly}(d)$ samples). We thus require (axis-wise) higher moments to be bounded. For the algorithm as described in this paper, these higher moments are also necessary for the deterministic conditions to hold. However, we believe additional pre-processing techniques such as axis-wise truncation can be used to remove this additional assumption. We emphasize again that this is beyond the scope of our present work.

---

### Official Review · Reviewer_AakN · 2022-07-11

**Rating:** 7
**Confidence:** 4
**Soundness:** 3 good
**Presentation:** 3 good
**Contribution:** 3 good

**Summary:**

This paper studies the problem of list-decodable mean estimation in the sparse setting. For distributions that have certifiable bounded t-th moment in any k-sparse directions, it proposes an algorithm that achieves error of (1/alpha)^{O(1/t)} in a sample and computationally-efficient manner. For Gaussian distribution, it achieves error rate \sqrt{log(1/alpha)} with quasi-polynomial complexity. The work leverages a technique called the “difference of pairs filter”, which filters on the difference between pairs of points instead of the points themselves, and a rounding scheme that recovers a reasonably good estimation of the underlying distribution.


**Questions:**

The structure of the paper can be improved (see Q2). It would be better if the technical highlights (probably for each algorithm) can be placed more evenly into the text, thus leaving the overview more concise. The paper can add more discussions on the merit of the “difference of pairs” technique and probably any other problems that could benefit from this new technique.


**Limitations:**

The work seems not have any negative societal impact as it is purely theoretical.


**Strengths And Weaknesses:**

Strengthes: The problem of list-decodable learning has addressed a flurry of recent interest in the machine learning community. It concerns learning from modern data that is usually corrupted with a high amount of noise, thus is considered to be significant and relevant. The proposed technique “DoP filter” is quite novel, which elegantly leverages the fact that the differences between the good points are close to zero. The performance of the algorithms are near-optimal.

Weaknesses: The structure and the presentation of the paper can be better. For example, the introduction seems to take up to 4.5 pages of the main text, while the algorithms, theorems, and proofs take the rest, making it harder to appreciate the contribution of this work.

---

> ### Author Response · Authors · 2022-08-02
> **Authors' response to Reviewer AakN**
>
> We thank the reviewer for their effort and positive assessment of our work. We refer to the joint response for the question about the merit of the difference-of-pairs and answer their specific question below:
>
> + **(Structure of the paper)** Thank you for your suggestion. We note that the introduction takes 4.5 pages, because we provide a comprehensive description of our new algorithmic technique in Section 1.3 of the introduction.  As per the reviewer’s request, we will add more prose and intuition to Section 3 in the revised version of the paper. This will be possible as an additional content page is allowed in the camera-ready version. We would like to emphasize that most of our paper (proofs included) fits into the main body, which is a testament to the simplicity of our framework.

---

> > ### Comment · Reviewer_AakN · 2022-08-07
> > **Raising my score**
> >
> > Thanks very much for the response. It would be helpful if more discussions can be added to Sec 3. On the other side, I agree with the authors that presenting most of the proofs in the main text also shows the simplicity of DoP framework. Thus, I am raising my score.

---

### Official Review · Reviewer_EBeb · 2022-07-11

**Rating:** 8
**Confidence:** 4
**Soundness:** 4 excellent
**Presentation:** 4 excellent
**Contribution:** 4 excellent

**Summary:**

This work considers list-decodable mean estimation: a small fraction (say 10%) of the data are iid draw from D, while the rest are adversarial. The goal is to output a finite number of estimates such that one of them approximates the mean of D. This had been broadly studied since [CSV17]. The main contribution of the current work is two-folds: 1) consider the sample complexity when the mean of D is sparse; and 2) design a new algorithm that runs in manageable time (in some regimes, polynomial time).

Overall, I think this is a solid work.

**Questions:**

See above.

**Ethics Review Area:**

["I don’t know"]

**Strengths And Weaknesses:**


1. Understanding the sample complexity of sparse mean estimation is important, and the work presents an algorithm with sample size poly(k, log(n)). This significantly improves prior results which attained poly(n) (though without the sparsity condition).

2. The use of sparsity in SoS system is interesting.

3. The developed difference-of-pairs filtering algorithm looks interesting, which did not appear in the literature. That being said, it is unclear whether such technique is vital to tackling the underlying problem under bounded moment condition. For example, a concurrent work shows similar results under Gaussian distribution without the new filtering approach:

- List-Decodable Sparse Mean Estimation, Zeng & Shen, arXiv:2205.14337

4. An SQ lower bound is also established and is useful to understand the fundamental limits of statistical estimation. I think it will be helpful to expand a bit on technical innovation to derive such SQ lower bound. How does it differ from [DKS17]?

5. While the SQ-type lower bound has been widely studied, it is weaker than information-theoretic bound. Can the authors comment on the challenge of characterizing the latter?

6. The description on sum-of-squares proof is rather short. I suggest authors include more background to make the paper self-contained. In particular, I saw that SoS is a useful tool to certify bounded moments, but is this the only purpose? Does it imply that for *any* general problems, SoS can be used in such a way as certification?

---

> ### Author Response · Authors · 2022-08-02
> **Authors' Response to reviewer EBeb**
>
> We thank the reviewer for their effort and positive assessment of our work. We refer to the joint response for the question about the merit of the difference-of-pairs (DoP) and answer their other questions below:
>
> ---
> 1. **(Relation of our SQ lower bounds with ones of prior work)** We follow the framework developed by [DKS17]. Our SQ lower bound is derived by modifying the instance of [DKS18a] to be sparse. In particular, the one-dimensional distribution $A$ used in the hard family that we construct is from [DKS18a], but instead of rotating that distribution according to a set of $2^{n^{\Omega(1)}}$ nearly-orthogonal (dense) unit vectors in $\mathbb{R}^d$, we replace that set by a set of $k^{n^{\Omega(1)}}$  nearly-orthogonal $k$-sparse unit vectors. We emphasize that while the SQ lower bound demonstrates that our algorithm achieves near-optimal guarantees for the problem, the main technical contribution of our paper is the novel DoP algorithm.
> ---
> 2. **(SQ bounds vs Information-theoretic bounds)** We respond to this question below by focusing our attention on the “Gaussian” setting.
>     + **(Information-theoretic sample complexity)** We remark that the sample complexity of list-decodable sparse estimation is relatively well-understood (see Footnote 3 on Page 17). In particular, there is a computationally-inefficient (i.e., with runtime exponential in the dimension) algorithm that, given a $(1-\alpha)$ corrupted set $T$ with $|T| \gg (k \log n)/\alpha^3$, obtains the minimax-optimal error of  $\sqrt{\log(1/\alpha)}$. It is easy to show that even when the identity of outliers are known, $\tilde{\Omega}((k \log n)/\alpha)$ samples are necessary.
>     + **(SQ lower bounds)** Informally speaking, SQ lower bounds are used to understand the “sample complexity” of “computationally-efficient” algorithms. Our SQ lower bound states that if we restrict ourselves to polynomial-time algorithms (formally, polynomial-time SQ algorithms), then the “sample complexity” to achieve the minimax-optimal error has to be at least $k^{\Omega \left( \frac{\log(1/\alpha)}{\log\log(1/\alpha)}\right)}$; this follows by setting $t = \Theta\left( \frac{\log(1/\alpha)}{\log\log(1/\alpha)}\right)$ in Theorem 1.3.
>     + **(Relation between the two bounds)** For list-decodable learning of Gaussians, the above two points show that there is a significant gap between the information-theoretic sample complexity and the SQ complexity. This is an instance of “information-computation gap” that has been recently shown for various problems (see the recent Simons program on this topic: https://simons.berkeley.edu/programs/si2021).
>
>       Both of these bounds serve different purposes, and they shed light on two different complexities of the statistical task.  We thus do not think that one of them is “weaker” than the other. Admittedly, the SQ lower bound is not the main contribution of our paper; the main contribution of our work is the novel DoP algorithm. Please let us know if we have misunderstood your question.
> ---
> 3. **(Background on SoS)** Due to space limitations, our description of the SoS system in the main body is succinct. We have provided additional relevant details about SoS in the Appendix, which is a standard practice in all SoS papers in the literature we are aware of. As per reviewer’s request, we will expand the background on SoS in the Appendix to clarify the purposes of the basic definitions and facts.
>
>     In general, SoS is useful when some polynomial relationship in the input needs to be exploited. These polynomials need not be moments. For instance, it can be used to certify anti-concentration of distributions [KKK19].
> ---

---

> > ### Comment · Reviewer_EBeb · 2022-08-03
> > **reviewer response**
> >
> > Thanks for the clarification. I suggest adding it to the revision. My rating is unchanged.

---

### Author Response · Authors · 2022-08-02
**Overall Response**

We thank the reviewers for their time and effort in providing feedback. We are encouraged by the universally positive scores, and that all the reviewers appreciated the paper for the following: (i) **significance** (EBeb, Ymm8), (ii) **novelty** (EBeb, AakN,Ymm8), (iii) **elegance and simplicity** (AakN,Ymm8).


We begin by addressing a question raised jointly by reviewers EBeb and AakN and proceed to answering individual questions later on. The reviewers asked whether the DoP technique is necessary for obtaining these guarantees, which we address below.

**(Benefits of the Difference-of-Pairs Filter)** Our new approach comes with several benefits that we list below:

- The Difference-of-Pairs (DoP) framework that we develop is conceptually simpler than prior list-decoding techniques that get an error of $o(1/\sqrt{\alpha})$, resulting in a cleaner algorithm with a compact analysis and a faster runtime. In particular, the algorithm and (most of its) analysis fit into the NeurIPS page limit, unlike the prior list-decoding algorithms. $\qquad\qquad\qquad\qquad\qquad\qquad\qquad\qquad\qquad\qquad\qquad\qquad\qquad\qquad\qquad\qquad\qquad\qquad\qquad\qquad\qquad\qquad\qquad\qquad\qquad$

- We now compare our approach with the prior techniques in the literature for list-decodable (dense) mean estimation that obtain error $o(1/\sqrt{\alpha})$:

   1. (SoS based algorithms of [KS17,RY19]) As we mention in the paper, these approaches are arguably quite complicated and have significantly worse runtime. While it may be possible to extend these approaches to the sparse setting after sufficient effort, our proposed algorithm easily adapts to sparsity (and possibly other ‘certifiable’ constraints).  $\qquad\qquad\qquad\qquad\qquad\qquad\qquad\qquad\qquad\qquad\qquad\qquad\qquad\qquad\qquad\qquad\qquad\qquad\qquad\qquad\qquad\qquad\qquad\qquad\qquad$

   2. (Multifilter-technique of [DKS18a]) [DKS18a] (and the concurrent work of Zeng and Shen) focus on the special case of Gaussian with identity covariance. These techniques do not extend to the more general case of log-concave (or certifiable distributions) or even Gaussians with unknown and bounded covariance. The concurrent work of Zeng and Shen extends the multifilter approach of [DKS18a] to the sparse setting. Even for the identity covariance Gaussian setting, the error guarantee of their algorithm gets stuck at  $\alpha^{-1/2}$. In contrast, our algorithm achieves a rate of $\alpha^{-1/t}$ in time $\text{poly}(mn^t)$ for a wider class of distributions for any $t$. In particular, our algorithm achieves the optimal error rate for identity covariance Gaussians when $t \approx \sqrt{\log(1/\alpha)}$. $\qquad\qquad\qquad\qquad\qquad\qquad\qquad\qquad\qquad\qquad\qquad\qquad\qquad\qquad\qquad\qquad\qquad\qquad\qquad\qquad\qquad\qquad\qquad\qquad\qquad$

  To conclude, while the DoP technique is perhaps not absolutely necessary to achieve the results we achieve (as mentioned above), it is far simpler than adapting existing techniques for the problem. By proposing a simple and novel framework for list-decoding mean estimation, we achieve both simplicity and generality.

---

### Meta-Review · Area_Chair_qucR · 2022-08-23

**Recommendation:** Accept
**Confidence:** Certain

**Metareview:**

This work considers list-decodable mean estimation: a small fraction (say 10%) of the data are iid draw from D, while the rest are adversarial. The goal is to output a finite number of estimates such that one of them approximates the mean of D. This had been broadly studied since [CSV17]. The main contribution of the current work is two-folds: 1) consider the sample complexity when the mean of D is sparse; and 2) design a new algorithm that runs in manageable time (in some regimes, polynomial time).

The referees are unanimous in recommending acceptance. The review of Ymm8 had to be disregarded due to a COI.

**Award:**

No

---

### Decision · Program_Chairs · 2022-09-14

Accept